# Complexes of vertebrate TMC1/2 and CIB2/3 proteins form hair-cell mechanotransduction cation channels

Arnaud PJ Giese[1†], Wei-Hsiang Weng[2,3†], Katie S Kindt[4], Hui Ho Vanessa Chang[5], Jonathan S Montgomery[2,6], Evan M Ratzan[7], Alisha J Beirl[4], Roberto Aponte Rivera[4], Jeffrey M Lotthammer[2], Sanket Walujkar[2], Mark P Foster[2,3,6], Omid A Zobeiri[8], Jeffrey R Holt[7], Saima Riazuddin[1], Kathleen E Cullen[9], Marcos Sotomayor[2,3,6*‡], Zubair M Ahmed[1,10,11*]

[1]Department of Otorhinolaryngology - Head & Neck Surgery, University of Maryland School of Medicine, Baltimore, United States; [2]Department of Chemistry and Biochemistry, The Ohio State University, Columbus, United States; [3]Biophysics Graduate Program, The Ohio State University, Columbus, United States; [4]Section on Sensory Cell Development and Function, National Institute on Deafness and Other Communication Disorders, National Institutes of Health, Bethesda, United States; [5]Department of Physiology, McGill University, Montreal, Canada; [6]Ohio State Biochemistry Program, The Ohio State University, Columbus, United States; [7]Departments of Otolaryngology and Neurology, F.M. Kirby Neurobiology Center, Boston Children's Hospital, Harvard Medical School, Boston, United States; [8]Department of Biomedical Engineering, McGill University, Montreal, Canada; [9]Departments of Biomedical Engineering, Neuroscience, and Otolaryngology and Head and Neck Surgery, Johns Hopkins University School of Medicine, Baltimore, United States; [10]Department of Biochemistry and Molecular Biology, University of Maryland School of Medicine, Baltimore, United States; [11]Department of Ophthalmology and Visual Sciences, University of Maryland School of Medicine, Baltimore, United States

*For correspondence:
sotomayor@uchicago.edu (MS);
zmahmed@som.umaryland.edu
(ZMA)

†These authors contributed
equally to this work

Present address: ‡Department
of Biochemistry and Molecular
Biology, University of Chicago,
Chicago, United States

Competing interest: The authors
declare that no competing
interests exist.

Reviewing Editor: Catherine
Emily Carr, University of
Maryland, United States

## eLife Assessment

This paper, on the role of calcium and integrin-binding protein 2 and 3 in the hair-cell in the mechano-electrical transduction (MET) apparatus, is a mix of confirmatory studies with new and potentially **important** data. Some parts, such as zebrafish studies, the modelling and simulations, are regarded as necessary and **convincing**. Other parts of the paper do not have the same novelty. Both Liang et al. (2021) and Wang et al. (2023) had previously demonstrated a role for CIB2/CIB3 in auditory and vestibular cells in mice. Moreover, there are also data in Riazuddin et al. (2012) paper that demonstrates the importance of CIB2 in zebrafish and *Drosophila*. Breaking the manuscript up to focus on specific aspects of the problem might alleviate the limitations of this multi-faceted study.

**Abstract** Calcium and integrin-binding protein 2 (CIB2) and CIB3 bind to transmembrane channel-like 1 (TMC1) and TMC2, the pore-forming subunits of the inner-ear mechano-electrical transduction (MET) apparatus. These interactions have been proposed to be functionally relevant across mechanosensory organs and vertebrate species. Here, we show that both CIB2 and CIB3 can form heteromeric complexes with TMC1 and TMC2 and are integral for MET function in mouse

cochlea and vestibular end organs as well as in zebrafish inner ear and lateral line. Our AlphaFold 2 models suggest that vertebrate CIB proteins can simultaneously interact with at least two cytoplasmic domains of TMC1 and TMC2 as validated using nuclear magnetic resonance spectroscopy of TMC1 fragments interacting with CIB2 and CIB3. Molecular dynamics simulations of TMC1/2 complexes with CIB2/3 predict that TMCs are structurally stabilized by CIB proteins to form cation channels. Overall, our work demonstrates that intact CIB2/3 and TMC1/2 complexes are integral to hair-cell MET function in vertebrate mechanosensory epithelia.

## Introduction

The vertebrate senses of hearing and balance depend on the process of MET in inner-ear hair cells. This process is initiated when mechanical forces from sound waves and head movements deflect specialized mechanosensory projections at the apical surface of hair cells, known as stereocilia (reviewed in *Fettiplace, 2017*; *Frolenkov et al., 2004*; *Gillespie and Müller, 2009*). Deflections of stereocilia result in tensioning of the extracellular tip links that convey mechanical force to the MET channels. Mature tip links are heterotetrameric filaments composed of cadherin 23 (CDH23) and protocadherin 15 (PCDH15) (*Ahmed et al., 2006*; *Kazmierczak et al., 2007*; *Siemens et al., 2004*; *Söllner et al., 2004*; *Sotomayor et al., 2012*). In mammalian auditory hair cells, MET channels are localized at the PCDH15 end of the tip link (*Beurg et al., 2009*). Significant progress in our understanding of MET has been recently made by identifying the transmembrane channel-like proteins 1 and 2 (TMC1 and TMC2) as the pore-forming subunits of the MET channel (*Kawashima et al., 2011*; *Pan et al., 2018*). In hair cells, two other transmembrane proteins, LHFPL5 and TMIE, are essential for sustained hearing. LHFPL5 binds to PCDH15 and may regulate force transmission to TMC1 and TMC2 (*Qiu et al., 2023*; *Xiong et al., 2012*). TMIE also interacts with TMC1/2 and regulates unitary conductance and ion selectivity of the MET channels (*Cunningham et al., 2020*; *Naz et al., 2002*). Interestingly, MET currents can still be recorded from hair cells lacking LHFPL5 or TMIE, suggesting that neither are essential for channel function (*Fettiplace et al., 2022*), yet all are needed for sustained hair-cell transduction and normal hearing. In zebrafish, Tmcs, Lhfpl5, Tmie, and Pcdh15 are also essential for sensory transduction, suggesting that these molecules form the core MET complex in all vertebrate hair cells (*Chen et al., 2020*; *Erickson et al., 2019*; *Erickson et al., 2017*; *Ernest et al., 2000*; *Gleason et al., 2009*; *Gopal et al., 2015*; *Maeda et al., 2017*; *Maeda et al., 2014*; *Pacentine and Nicolson, 2019*; *Phillips et al., 2011*; *Seiler et al., 2004*; *Söllner et al., 2004*).

We previously showed that calcium ($Ca^{2+}$) and integrin-binding protein 2 (CIB2), and the closely related CIB3, interact with TMC1/2 (*Giese et al., 2017*). Furthermore, we also demonstrated that CIB2 is essential for MET function in auditory hair cells (*Giese et al., 2017*). Recessive variants of *CIB2* predominantly cause nonsyndromic DFNB48 hearing loss in humans (*Booth et al., 2018*; *Michel et al., 2017*; *Riazuddin et al., 2012*; *Seco et al., 2016*; *Wang et al., 2017*). Similarly, recessive alleles of *Cib2* cause hearing loss in mice but no vestibular deficits (*Giese et al., 2017*; *Michel et al., 2017*; *Wang et al., 2017*). In zebrafish, the knockdown of *Cib2* diminishes both the acoustic startle response and mechanosensitive responses of lateral-line hair cells (*Riazuddin et al., 2012*). CIB2 belongs to a family of four closely related proteins (CIB1-4) that have partial functional redundancy and similar structural domains, with at least two $Ca^{2+}/Mg^{2+}$-binding EF-hand motifs that are highly conserved for CIB2/3 (*Huang et al., 2012*). Wang and co-authors investigated the function of *Cib1* in cochlear hair cells through the generation of a knockout strain and concluded that although *Cib1* is expressed in the mouse cochlear hair cells, loss of CIB1 protein did not affect auditory function (*Wang et al., 2017*). A recent study demonstrated that while CIB2 is essential for the proper localization of TMC1/2 at the tips of auditory hair cell stereocilia, CIB3 can restore MET currents in cochlear hair cells lacking CIB2 (*Liang et al., 2021*). The same study reported a crystal structure of a TMC1 intracellular linker in complex with CIB3, which suggests that CIB2/3 proteins are likely to be auxiliary subunits of hair cell MET channels (*Liang et al., 2021*).

Previous work on CIB proteins primarily focused on auditory hair cells, leaving open the question of whether they play a role in vestibular function. This question was addressed in part by another recent study that reported vestibular deficits in mice lacking both CIB2 and CIB3 (*Wang et al., 2023*). However, additional evidence showing that CIB2/3 can function and interact with TMC1/2 proteins across sensory organs, hair-cell types, and species is still needed. Here, we provide further evidence

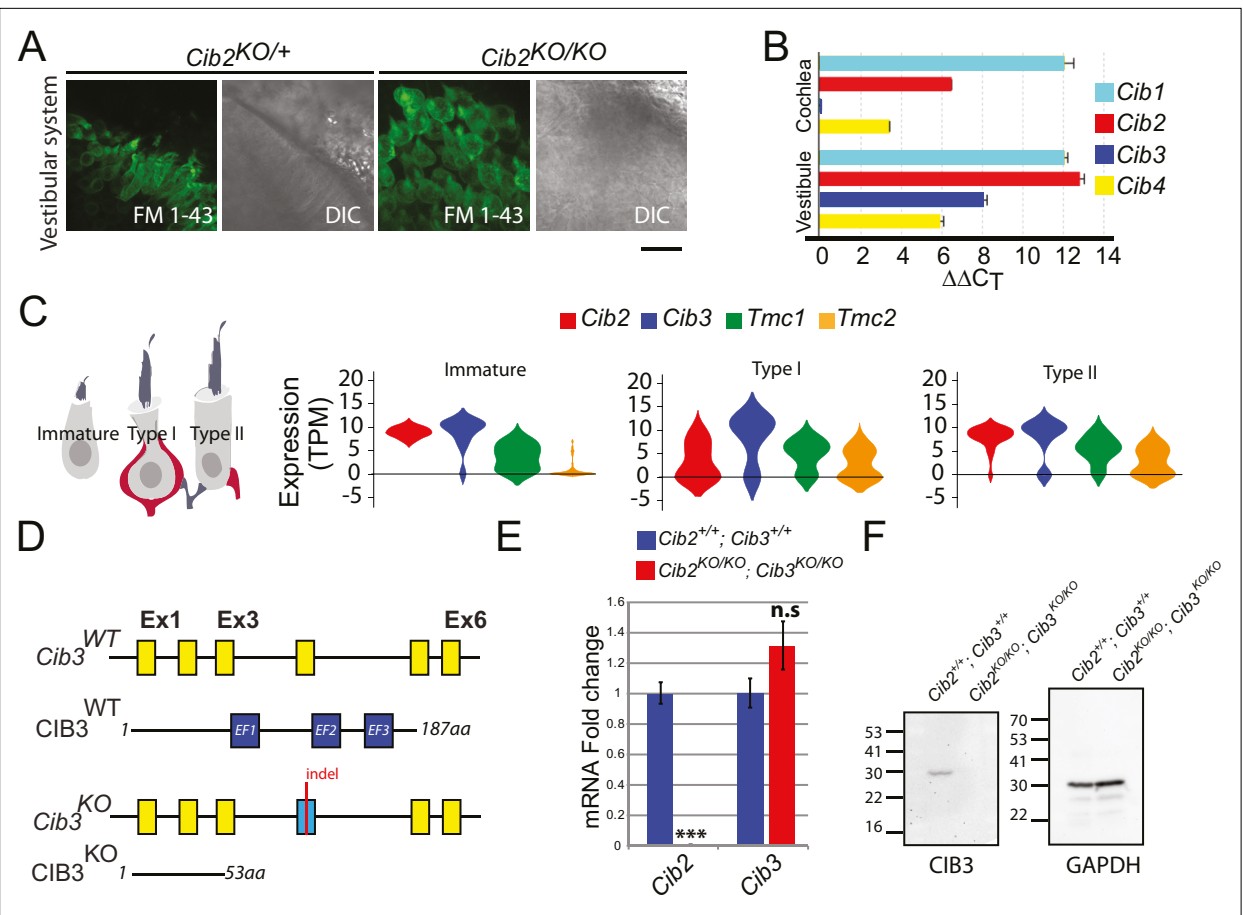

**Figure 1.** Expression of *Cib* genes in the mouse inner ear and generation of *Cib3* knockout mice. (**A**) Maximum intensity projections of Z-stacks of confocal fluorescent images and corresponding DIC images of *Cib2*^KO/+ and *Cib2*^KO/KO cultured vestibular end organ explants imaged after exposure to 3 µM of FM 1–43 for 10 s. The samples were dissected at P5 and kept 2 d in vitro (P5 + 2 div). Scale bar: 20 µm. (**B**) Real-time quantitative RT–PCR analysis of mRNA levels in the cochlear and vestibular sensory epithelia at P12 revealed different expressions of *Cib1, Cib2, Cib3,* and *Cib4* in the auditory and vestibular end organs. (**C**) Violin plots generated by gEAR portal showing the expression levels of *Cib2, Cib3, Tmc1,* and *Tmc2* by scRNA-seq in immature, type I, and type II utricular vestibular hair cells. TPM: transcript per million. (**D**) Gene structure of the wild-type and *Cib3*^KO alleles. Exons are shown in yellow. Indel in exon 4 leads to a nonsense mutation and a premature stop at the protein level (p.Asp54*) (**E**) Real-time quantitative RT–PCR analysis using cDNA from brain tissue of *Cib2;Cib3* mutant mice was used to analyze the expression of *Cib2* and *Cib3*. Contrary to *Cib2* mRNA, *Cib3* mRNA appears to be stable in the *Cib2*^KO/KO;*Cib3*^KO/KO mice and does not go through nonsense-mediated decay process. The relative expression of *Cib2* and *Cib3* members were normalized against *hprt*. Asterisks indicate statistical significance: ***p<0.001 (Student's t-test). n.s.: not significant. Error bars represent SEM. (**F**) Western Blot analysis on *Cib2;Cib3* mouse heart tissue was performed using CIB3 antibody. No CIB3 protein was detected in the *Cib2*^KO/KO;*Cib3*^KO/KO mice, validating our mouse model. GAPDH detection was used as a loading control.

The online version of this article includes the following source data for figure 1:

**Source data 1.** Raw data for plots of *cib2* and *cib3* expression; original image for western blot gel.

that CIB2 and CIB3 are indeed evolutionary conserved and functionally redundant components of the vertebrate transduction apparatus necessary for MET channel activity in mouse vestibular hair cells, as well as in zebrafish mechanosensory epithelia. Moreover, based on our structural analyses and simulations, we propose that CIB2 and CIB3 are required for the stabilization and channel function of the MET proteins TMC1 and TMC2.

# Results

## CIB2 and CIB3 are expressed in mouse vestibular hair cells

We and others previously demonstrated that CIB2 is essential for MET in auditory hair cells (*Giese et al., 2017*; *Michel et al., 2017*; *Wang et al., 2017*). In contrast to auditory hair cells, we found that

the vestibular hair cells in $Cib2^{KO/KO}$ mice apparently have MET. We assessed MET via uptake of FM 1–43 (*Figure 1A*), a styryl dye that mostly permeates into hair cells through functional MET channels (*Meyers et al., 2003*), indicating that there may be another CIB protein playing a functionally redundant role. Bulk RNA sequencing revealed that all four members of the *Cib* family are expressed in mouse inner ear tissues (*Figure 1B*). However, when assessed by single-cell RNA sequencing analysis, only *Cib2* and *Cib3* are expressed in immature hair cells in the mouse utricle (*Figure 1C*), a vestibular sensory organ required to detect linear acceleration (*McInturff et al., 2018*). Furthermore, we find that the expression of *Cib2* and *Cib3* is maintained in mature type I and type II utricular hair cells (*Figure 1C*). We also examined the expression of *Tmc1* and *Tmc2* in utricular hair cells and found that in immature hair cells, *Tmc1* is highly expressed, while *Tmc2* expression is low (*Figure 1C*). *Tmc2* expression increases during development but remains below *Tmc1* levels in both type I and type II hair cells upon maturation (*Figure 1C*). CIB2 and CIB3 have been reported to interact with TMC1/2, and CIB2 is essential for auditory hair cell MET (*Giese et al., 2017*; *Liang et al., 2021*). The co-expression of *Cib2* and *Cib3* in developing and mature vestibular hair bundles, and the ability of CIB2/3 to interact with TMC1 and TMC2, all make them attractive candidates to regulate MET function in vestibular hair cells.

## CIB3 is not required for hearing function in young mice

To evaluate whether CIB3 regulates MET function both in auditory and vestibular hair cells, we generated *Cib3* knockout (*Cib3^KO*) mice using CRISPR/Cas9 technology (*Figure 1D*). An indel in exon 4 of *Cib3* was introduced leading to a nonsense mutation (p.Asp54*), thus causing a loss of allele function (*Figure 1E–F*). We first characterized hearing function in $Cib3^{KO/KO}$ and control littermate mice at P16 by measuring auditory-evoked brainstem responses (ABRs). Normal ABR waveforms and thresholds were observed in $Cib3^{KO/KO}$ indicating normal hearing. Next, we investigated if there was a functional redundancy between CIB2 and CIB3 in auditory hair cells. For this analysis, the $Cib3^{KO}$ allele was introduced into the $Cib2^{KO}$ background. Similar to what has been previously reported in $Cib2^{KO/KO}$ mice (*Giese et al., 2017*; *Michel et al., 2017*; *Wang et al., 2017*), the *Cib2:Cib3* double knockout ($Cib2^{KO/KO};Cib3^{KO/KO}$) mice were also profoundly deaf. The $Cib2^{KO/KO};Cib3^{KO/KO}$ mice did not respond to click or strong tone burst stimuli at 100 dB sound pressure level (SPL) (*Figure 2A, B*). As reported previously for $Cib2^{KO/KO}$ mice, we observed that auditory hair cells in $Cib2^{KO/KO};Cib3^{KO/KO}$ mice did not accumulate FM 1–43 dye (*Figure 2C*). Also, similar to $Cib2^{KO/KO}$ mice (*Giese et al., 2017*; *Michel et al., 2017*; *Wang et al., 2017*), we observed progressive degeneration of auditory hair cells in $Cib2^{KO/KO};Cib3^{KO/KO}$ mice (*Figure 2D*). Taken together with previous studies, our results confirm that CIB2 is indispensable for hearing and MET function in the auditory hair cells.

## CIB2 and CIB3 are required for vestibular function in mice

After we assessed auditory function, we turned our analyses toward vestibular function. Similar to $Cib2^{KO/KO}$ mice (*Giese et al., 2017*), $Cib3^{KO/KO}$ mice did not display any obvious indications of vestibular dysfunction, such as circling, hyperactivity, or head bobbing, performed well when exploratory behavior was tested, and had normal vestibulo-ocular reflexes (*Figure 3*). In contrast, visual observation of the $Cib2^{KO/KO};Cib3^{KO/KO}$ mice indicated altered exploratory behavior in an open-field test, suggesting an impairment of the vestibular system (*Figure 3A–B*). To assess the implications of the $Cib2^{KO/KO};Cib3^{KO/KO}$ mice on vestibular function, we next performed a series of tests, including: the vestibulo-ocular reflex (VOR), performance in a rotating rod, and head movement analysis (*Figure 3C–H*). The VOR is mediated by a direct three-neuron pathway linking the vestibular nerve to the extraocular muscles and produces compensatory eye movements in the direction opposite of the head movement to provide a stable gaze (reviewed in *Cullen, 2019*). The dynamic response properties (i.e. gain and phase) of the VOR can be precisely quantified in response to rotations applied at different frequencies in darkness (VORd). In response to VORd testing, $Cib2^{+/+};Cib3^{KO/KO}$ mice generated robust compensatory eye movements (*Figure 3C*, black trace). In contrast, $Cib2^{KO/KO};Cib3^{KO/KO}$ mice generated negligible eye movements (*Figure 3C*, red trace). Consistent with this observation, quantification of the VORd across frequencies (*Figure 3D*) revealed gains near zero in the $Cib2^{KO/KO};Cib3^{KO/KO}$ mice that were significantly reduced (p<0.05) compared to those measured in $Cib2^{+/+};Cib3^{KO/KO}$ mice. Interestingly, the VORd gains of $Cib2^{KO/+};Cib3^{KO/KO}$ mice also showed attenuation with a significant decrease at 2 Hz (p<0.01) compared to $Cib2^{+/+};Cib3^{KO/KO}$ mice. On the other hand, the response phase of VORd

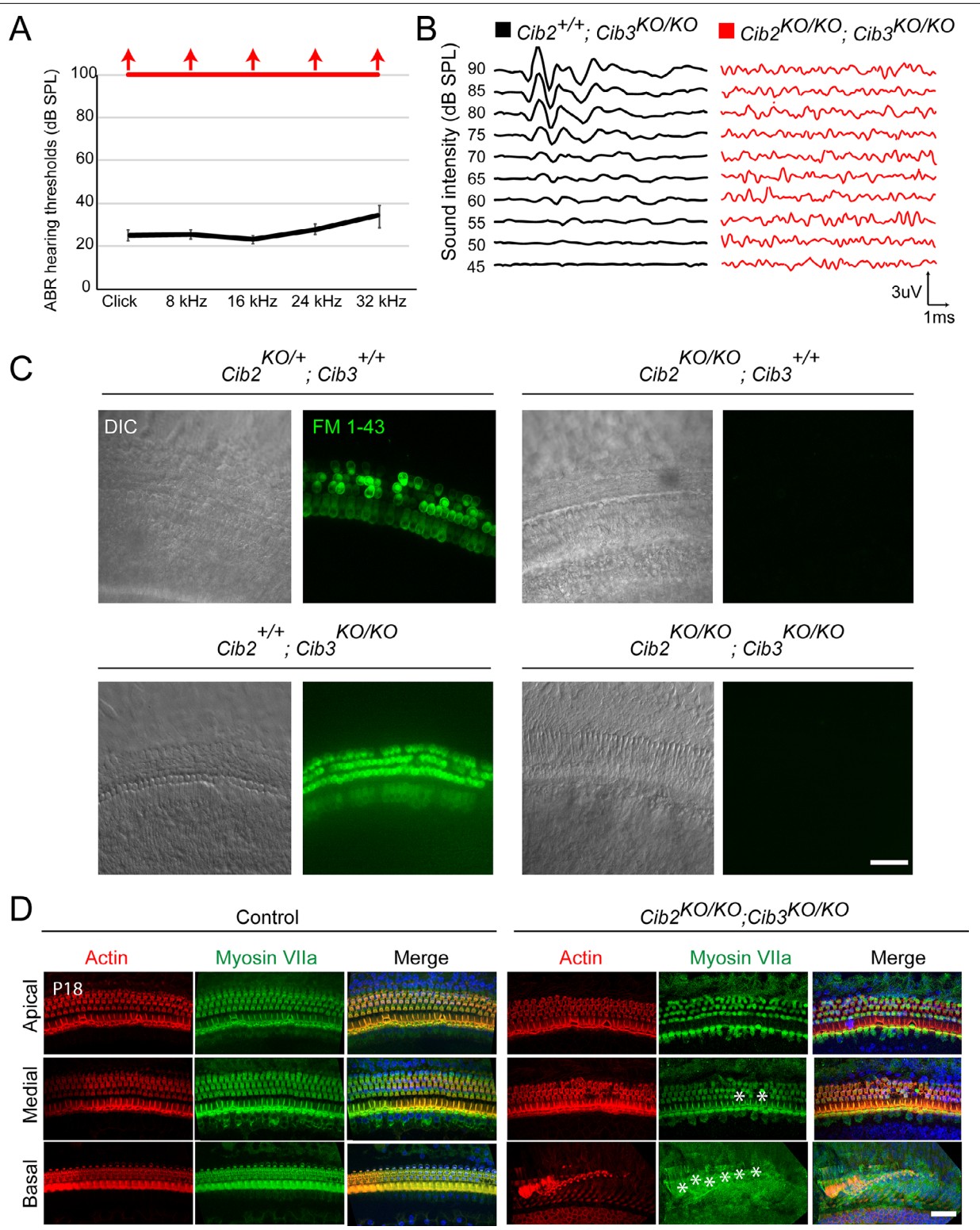

**Figure 2.** CIB2/3 double mutant mice have profound hearing loss. (**A**) Auditory-evoked brainstem response (ABR) thresholds to broadband clicks and tone-pips with frequencies of 8 kHz, 16 kHz, 24 kHz, and 32 kHz in $Cib2^{+/+};Cib3^{KO/KO}$ (black; n=5) and $Cib2^{KO/KO};Cib3^{KO/KO}$ (red, n=5) mice at P16. The same animals were tested with clicks and tone pips. (**B**) ABR pure tone traces of P32 $Cib2^{+/+};Cib3^{KO/KO}$ (black) and $Cib2^{KO/KO};Cib3^{KO/KO}$ (red) mice measured at 32 kHz. Hearing thresholds are in Decibels (dB SPL). (**C**) Maximum intensity projections of Z-stacks of confocal fluorescent images (right) and corresponding DIC images (left) of $Cib2^{KO/+};Cib3^{+/+}$, $Cib2^{KO/KO};Cib3^{+/+}$, $Cib2^{+/+};Cib3^{KO/KO}$ and $Cib2^{KO/KO};Cib3^{KO/KO}$ cultured organ of Corti explants imaged after exposure to 3 μM of FM 1–43 for 10 s. The samples were dissected at P5 and kept 2 d in vitro (P5 +2 div). Scale bar: 20 μm. (**D**) Maximum

*Figure 2 continued on next page*

*Figure 2 continued*

intensity projections of confocal Z-stacks of apical, medial, and basal turns of organ of Corti of control and *Cib2^KO/KO^;Cib3^KO/KO^* mice immunostained with an anti-myosin VIIa antibody (green) and counterstained with phalloidin (red) at P18. Asterisks indicate hair cell loss. Scale bar: 10 µm.

The online version of this article includes the following source data for figure 2:

**Source data 1.** Raw data for ABR plots.

eye movements was comparable for *Cib2^KO/+^;Cib3^KO/KO^* and *Cib2^+/+^;Cib3^KO/KO^* mice at all frequencies (p>0.05) (*Figure 3D*).

As a control for possible changes in the visual system and/or ocular motor pathway, we also quantified the optokinetic reflex (OKR), which generates eye movements in response to a moving visual scene and in light. The OKR normally works together with the VOR to stabilize the visual field on the retina. Both *Cib2^+/+^;Cib3^KO/KO^* and *Cib2^KO/+^;Cib3^KO/KO^* showed robust OKN responses which similarly decreased as a function of frequency (*Figure 3E*; p>0.05). Likewise, OKR gain values in *Cib2^KO/KO^;Cib3^KO/KO^* were comparable with the single exception of a significant gain decrease at 0.2 Hz (p<0.05). Analysis of OKN similarly revealed a comparable response phase for all three groups of mice at all frequencies (p>0.05) with the exception that the *Cib2^KO/KO^;Cib3^KO/KO^* mice demonstrated less lag at 0.8 and 2 Hz (*Figure 3E–F*). Overall, the OKR responses of mutant and control mice were generally comparable.

In addition to generating compensatory VOR eye movements to stabilize gaze, the vestibular system also plays a key role in the maintenance of posture and control of balance. Thus, we also tested all three groups of mice on several vestibular tasks that assessed balance and postural defects. First, quantification of open-field testing experiments confirmed that mice exhibited circling – a behavior that is commonly associated with loss of vestibular function in mice (*Figure 3B*; p<0.002). We also completed rotarod testing in which performance was quantified by measuring the amount of time the mouse remained on the rotating rod as its acceleration increased. While *Cib2^+/+^;Cib3^KO/KO^* and *Cib2^KO/+^;Cib3^KO/KO^* mice exhibited similar performance, *Cib2^KO/KO^;Cib3^KO/KO^* mice exhibited significantly poorer performance across all time points from day 1 through day 4 (*Figure 3G*, p<0.002). Finally, it was also possible to distinguish *Cib2^KO/KO^;Cib3^KO/KO^* mice from the other two groups in their home cages since they commonly displayed head bobbing, a trait also generally associated with vestibular dysfunction in mice. To quantify head movement in our mice, we attached a 6D MEMS module consisting of three gyroscopes and three linear accelerometers to the mouse's head implant (see Methods). Analysis of power spectra of head motion revealed significantly higher power for *Cib2^KO/KO^;Cib3^KO/KO^* mice in all six dimensions of motion, as compared to *Cib2^KO/+^;Cib3^KO/KO^* and *Cib2^+/+^;Cib3^KO/KO^* mice (*Figure 3H*, compare red vs. blue and black). Furthermore, there were no significant differences in the head movements power spectra of *Cib2^KO/+^;Cib3^KO/KO^* and *Cib2^+/+^;Cib3^KO/KO^* mice (*Figure 3H*, blue vs. black). Taken together our findings indicate that *Cib2^KO/KO^;Cib3^KO/KO^* mice show marked deficits in vestibulomotor function when compared to both *Cib2^KO/+^;Cib3^KO/KO^* and *Cib2^+/+^;Cib3^KO/KO^* mice.

## CIB2 and CIB3 are functional subunits of the MET apparatus in mouse vestibular hair cells

To understand the root cause of vestibulomotor function deficits described above for *Cib2^KO/KO^;Cib3^KO/KO^* mice, we first analyzed vestibular hair bundle morphology using confocal microscopy. High-resolution confocal imaging did not reveal any obvious vestibular hair cell loss in *Cib2^KO/KO^;Cib3^KO/KO^* mice (*Figure 4A*). Next, we visualized MET activity in vestibular hair cells of saccular and utricular end organs of *Cib2^KO/KO^;Cib3^KO/KO^* mice using the styryl dye FM 1–43. These vestibular end organs are divided into two main physically and functionally distinct domains, a central striolar domain and a peripheral extrastriolar domain. At P5, FM 1–43 labeling of *Cib2^KO/KO^; Cib3^KO/KO^* tissue failed to label vestibular hair cells in appreciable numbers (*Figure 4B*). Similarly, fixable FM 1–43 FX intraperitoneal injections at P60 also showed no dye uptake in *Cib2^KO/KO^; Cib3^KO/KO^* mice (*Figure 4C*). *Cib2^KO/KO^* and *Cib3^KO/KO^* single mutant mice had dye uptake both in striolar and extrastriolar hair cells, although a subtle reduction in FM 1–43 FX was observed in extrastriolar hair cells of both single mutants at P60 (*Figure 4C*). Taken together, these results support compulsory but functionally redundant roles for CIB2 and CIB3 in the vestibular hair cell MET complex.

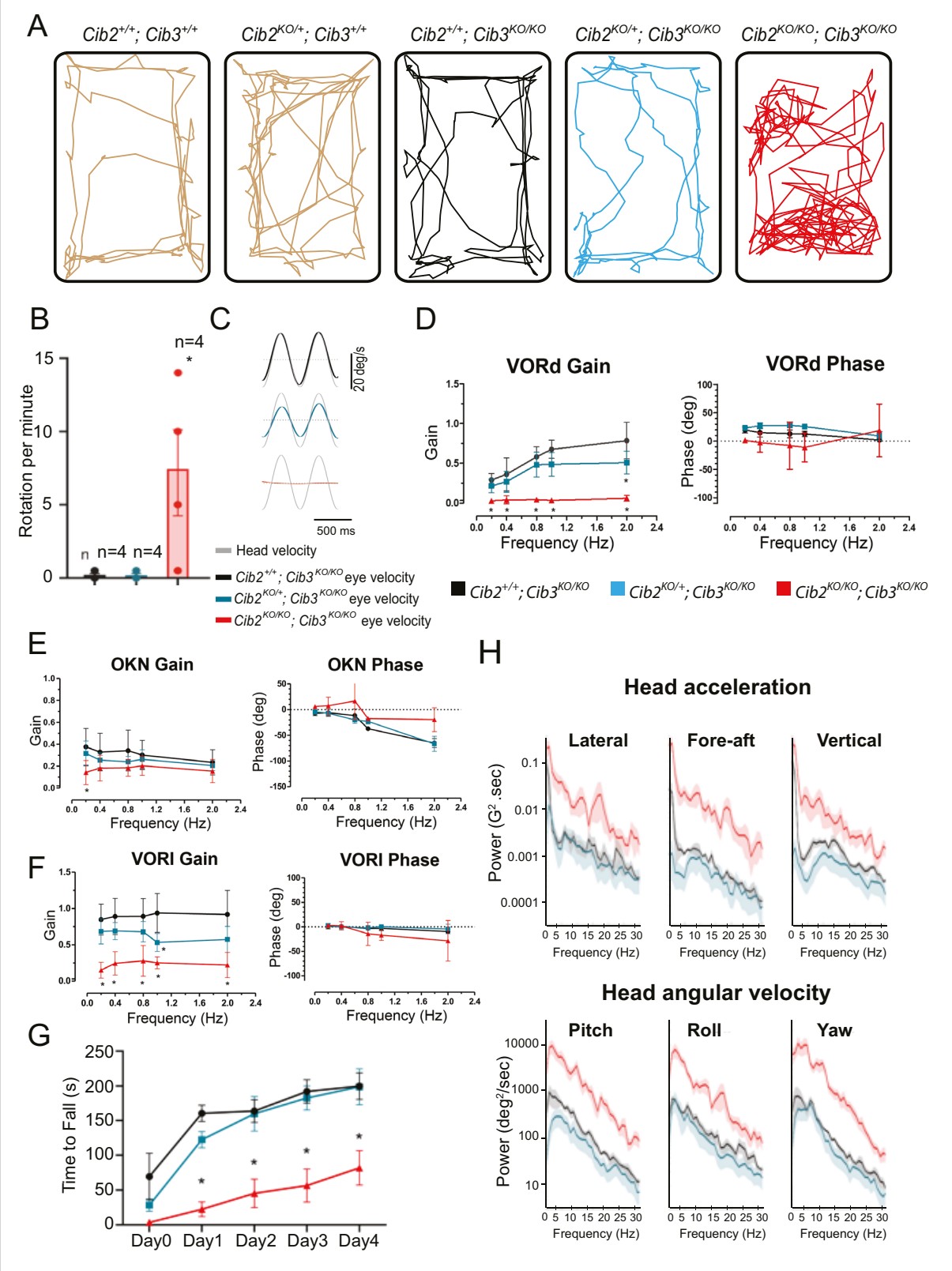

**Figure 3.** CIB2/3 double mutant mice have vestibular dysfunction. (**A**) Traces showing the open-field exploratory behavior of P60 *Cib2+/+;Cib3+/+*, *Cib2KO/+;Cib3+/+*, *Cib2+/+;Cib3KO/KO*, *Cib2KO/+;Cib3KO/KO* and *Cib2KO/KO;Cib3KO/KO* mutant mice. (**B**) Quantification of the number of rotations in 120 s (mean ± SEM), showing that, unlike *Cib2+/+;Cib3KO/KO*, and *Cib2KO/+;Cib3KO/KO* mice, *Cib2KO/KO;Cib3KO/KO* mutant mice display a circling behavior and a vestibular defect (t-test) (n=4 for each genotype). (**C**) Examples of head-velocity (gray) and resultant eye-velocity traces evoked during VORd testing. Note, that

*Figure 3 continued on next page*

Figure 3 continued

the eye movement is compensatory, and the trace has been inverted to facilitate comparison with head velocity. (**D**) VORd gain and phase (mean ± SD) plotted as a function of frequency for $Cib2^{+/+};Cib3^{KO/KO}$ (n=4), $Cib2^{KO/+};Cib3^{KO/KO}$ (n=5), $Cib2^{KO/KO};Cib3^{KO/KO}$ (n=4). (**E**) OKN gain and phase (mean ± SD) plotted as a function of frequency for $Cib2^{+/+};Cib3^{KO/KO}$ (n=4), $Cib2^{KO/+};Cib3^{KO/KO}$ (n=5), $Cib2^{KO/KO};Cib3^{KO/KO}$ (n=4). (**F**) VORl gain and phase (mean ± SD) plotted as a function of frequency for $Cib2^{+/+};Cib3^{KO/KO}$ (n=4), $Cib2^{KO/+};Cib3^{KO/KO}$ (n=5), $Cib2^{KO/KO};Cib3^{KO/KO}$ (n=4). Comparisons made with two-way ANOVA followed by Bonferroni post-hoc test for (**B,D-F**); *$p < 0.05$. (**G**) Quantification of the time mice remained on the rotating rod with increasing acceleration (mean ± SEM). Comparisons made with two-way ANOVA followed by Bonferroni post-hoc test ; *$p < 0.002$. ($Cib2^{+/+};Cib3^{KO/KO}$ (n=6), $Cib2^{KO/+};Cib3^{KO/KO}$ (n=6), $Cib2^{KO/KO};Cib3^{KO/KO}$ (n=6)). (**H**) Comparison of power spectra density of head movements in translational axes (top) and rotational axes (bottom) between wild type ($Cib2^{+/+};Cib3^{KO/KO}$, black), heterozygous mutants ($Cib2^{KO/+};Cib3^{KO/KO}$, blue), and homozygous mutants ($Cib2^{KO/KO};Cib3^{KO/KO}$, red). The double knockout $Cib2^{KO/KO};Cib3^{KO/KO}$ exhibit significantly higher power than $Cib2^{+/+};Cib3^{KO/KO}$ and $Cib2^{KO/+};Cib3^{KO/KO}$ across all frequencies (0–30 Hz), in all six translational and rotational axes.

The online version of this article includes the following source data for figure 3:

**Source data 1.** Raw data for plots of rotations, VORd, OKN, VORl, and time to fall; matlab code to convert and plot head movement data.

## Cib2 and Cib3 function is evolutionary conserved in zebrafish

Numerous studies have shown that the same proteins are required for hair cell MET in zebrafish and mammals (*Chen et al., 2020*; *Erickson et al., 2017*; *Ernest et al., 2000*; *Gleason et al., 2009*; *Gopal et al., 2015*; *Maeda et al., 2017*; *Maeda et al., 2014*; *Phillips et al., 2011*; *Seiler et al., 2004*; *Söllner et al., 2004*). Unlike mammals, the zebrafish's inner ear does not have a cochlea. Instead, zebrafish rely on their saccule for hearing and utricle for balance. Zebrafish also utilize hair cells in a specialized sensory system, the lateral line, to detect local water movement. The hair cells within both the zebrafish inner ear and lateral line share many morphological similarities to mammalian vestibular hair cells (*Nicolson, 2017*). Zebrafish scRNAseq data show that both *cib2* and *cib3* are expressed in the inner ear while primarily *cib2* is expressed in lateral-line hair cells (*Baek et al., 2022*; *Shi et al., 2023*). Given the zebrafish reliance on the saccule for hearing and the similarities between zebrafish hair cells to mammalian vestibular hair cells, we tested whether Cib2 and/or Cib3 were required for hair cell MET in the zebrafish inner ear or lateral line.

To examine the impact of Cib2 and Cib3 loss in zebrafish, we created mutants using CRISPR-Cas9-based methods to target each homologue (*Varshney et al., 2016*). Both mutants lead to a stop codon in the last EF-hand domain (*Figure 5—figure supplement 1*). We first examined auditory function by testing the acoustic startle reflex in *cib2* and *cib3* mutants. For our experiments, we used a Zantiks behavioral system to quantify the probability of generating a startle in response to a vibrational tap stimulus. This response is thought to rely primarily on hair cells in the zebrafish saccule and lateral-line system. We found that, compared to controls, the probability of a startle response was significantly reduced in *cib2* mutants (*Figure 5A*, control: 0.71 ± 0.04; *cib2*: 0.28 ± 0.06, p=0.005). In contrast, *cib3* mutants startle with a similar frequency compared to controls (*Figure 5A*, p=0.36). Interestingly, contrary to mammals, where CIB2 is crucial for hearing, acoustic startle responses were not completely eliminated in *cib2* mutants. One possibility for this partial loss of acoustic startle is that Cib3 may be able to compensate for Cib2 in the zebrafish. Therefore, we examined acoustic startle responses in *cib2;cib3* mutants. In *cib2;cib3* mutant animals we observed a complete absence of startle responses (*Figure 5A*). Furthermore, we observed that *cib2;cib3* mutants but not *cib2* or *cib3* mutants exhibited spontaneous circling behavior, an additional indicator of inner ear dysfunction in zebrafish (*Nicolson et al., 1998*). Together our behavioral results indicate that Cib2 and Cib3 are both required to ensure normal acoustic startle responses in zebrafish. Furthermore, just loss of Cib2, but not Cib3 can impair auditory function in zebrafish.

## Cib2 and Cib3 are required for MET in the zebrafish lateral line

Our results demonstrate that both Cib2 and Cib3 are required for proper auditory function in zebrafish. This raises the question of whether, similar to mice, these proteins are critical for MET function in zebrafish hair cells. Therefore, we next turned our analysis to hair cells in neuromast organs of the lateral line where methods to assess MET are well-documented and straightforward (*Lukasz and Kindt, 2018*; *Seiler et al., 2004*).

First, to assess whether there were gross alterations to MET in lateral-line hair cells we examined uptake of the FM 1–43. We found that, compared to controls, the percent of hair cells per neuromast labeling with FM 1–43 was significantly reduced in *cib2* mutants (*Figure 5B, D and E*, control: 97.6 ±

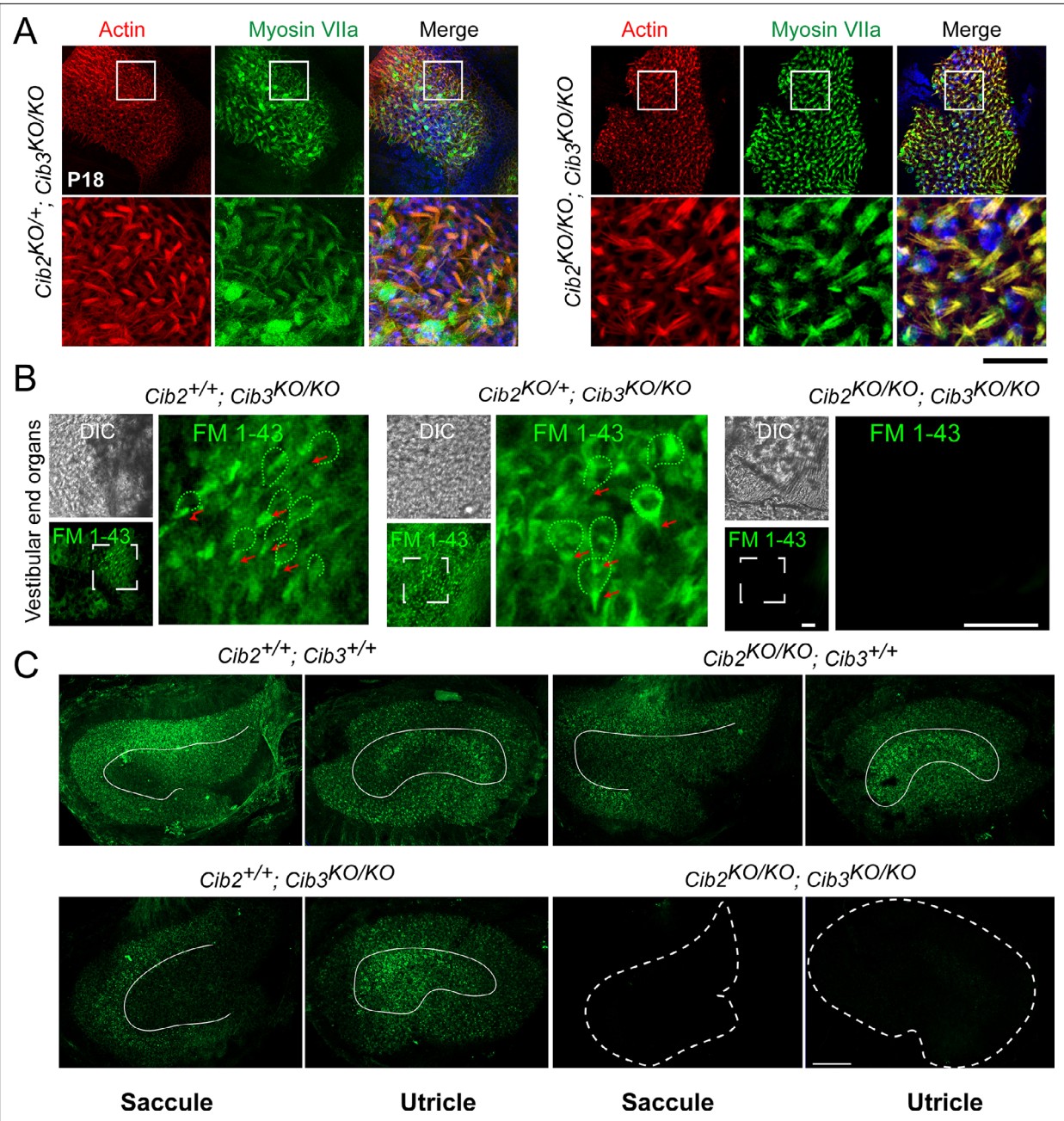

**Figure 4.** Vestibular hair cells do not have functional mechano-electrical transduction (MET) channels at rest in CIB2/3 double mutant mice. (**A**) Maximum intensity projections of confocal Z-stacks of the vestibular end organs of P18 $Cib2^{KO/+};Cib3^{KO/KO}$ and $Cib2^{KO/KO};Cib3^{KO/KO}$ mutant mice immunostained with myosin VIIa antibody (green) and counterstained with phalloidin (red) and DAPI (blue). Scale bar = 50 µm. (**B**) Adult vestibular end organs of $Cib2^{+/+};Cib3^{KO/KO}$, $Cib2^{KO/+};Cib3^{KO/KO}$, and $Cib2^{KO/KO};Cib3^{KO/KO}$ mutant mice imaged after exposure to 3 µM of FM 1–43 for 10 s. Vestibular hair cell bodies (green dotted lines), and stereocilia bundles (red arrows) are shown. Scale bar: 100 µm. (**C**) FM 1–43 FX (fixable form) labeling of saccules and utricles by IP injection at P60. Obvious reduction in signal in $Cib2^{KO/KO};Cib3^{KO/KO}$, and subtle extrastriolar reduction in $Cib2^{+/+};Cib3^{KO/KO}$ mice was observed. Extrastriolar regions are delimited with white lines. Scale bar = 100 µm.

1.2; $cib2$: 37.5 ± 6.9, p<0.0001). In contrast, a similar percent of hair cells per neuromast labeled with FM 1–43 in $cib3$ mutants, compared to controls (*Figure 5B, D and F*, $cib3$: 95.7 ± 1.9, p=1.0). We also quantified the average intensity of the FM 1–43 in hair cells (*Figure 5C*). We found that, compared to controls, the average FM 1–43 label was reduced in $cib2$ but not $cib3$ mutants (*Figure 5C*). In $cib2$ mutants we did observe residual hair cells that were labeled with FM 1–43. We hypothesized that in lateral-line hair cells, similar to its role in zebrafish auditory function, Cib3 may be able to substitute

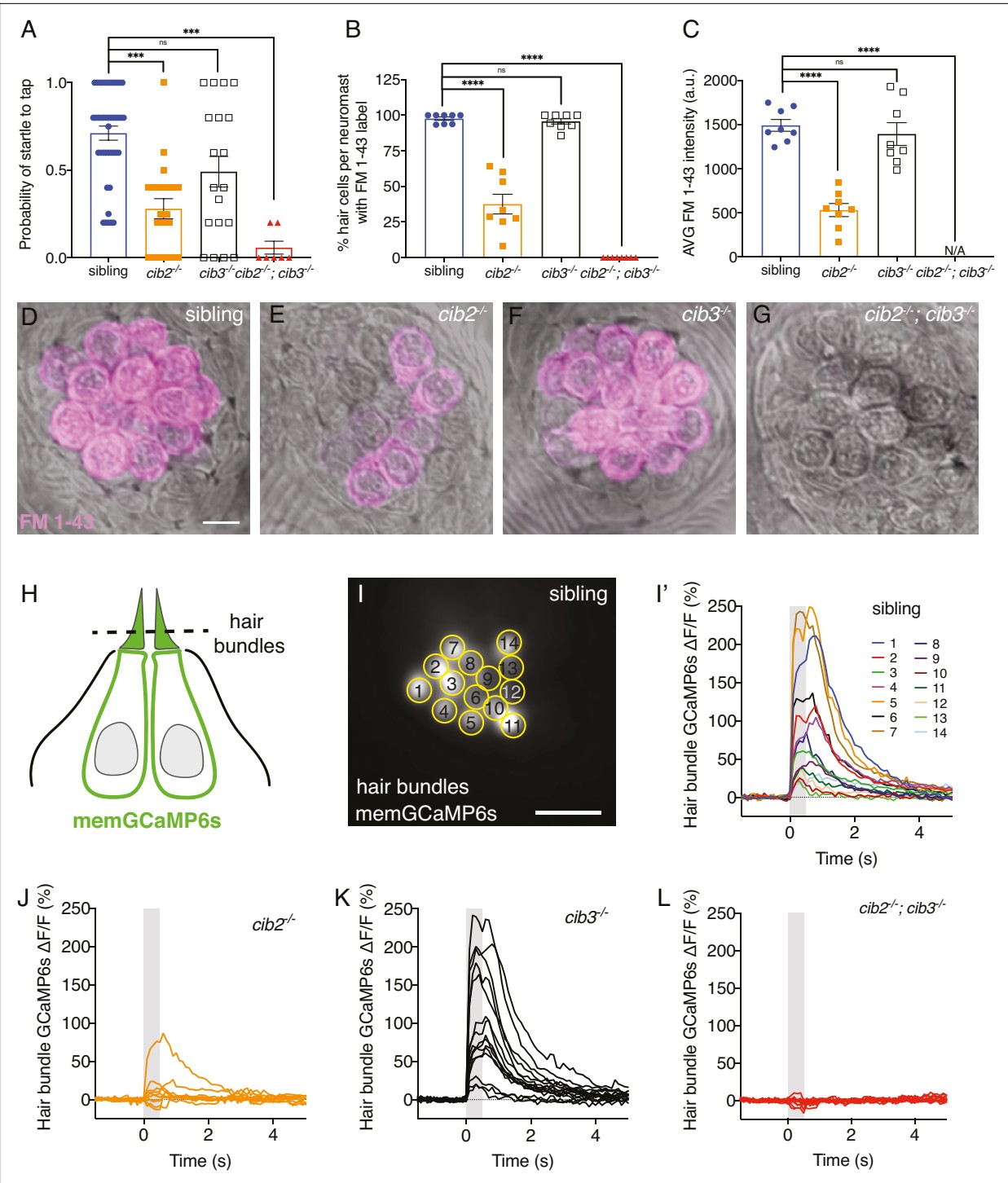

**Figure 5.** Both Cib2 and Cib3 are required in zebrafish for acoustic startle and mechano-electrical transduction (MET) function. (**A**) Compared to controls (siblings) *cib3* mutants have a normal acoustic startle response. *Cib2;cib3* double mutants completely lack an acoustic startle response, while *cib2* mutants have a reduced probability to startle compared to controls (n=43 sibling, 20 *cib2*, 19 *cib3*, and 7 *cib2;cib3* animals). (**B**) Compared to controls *cib3* mutants have a normal % of hair cells that label with FM 1–43 per neuromast. In *cib2;cib3* mutants no hair cells label with FM 1–43, and in *cib2* mutants a significantly reduced % of hair cells label with FM 1–43 per neuromast. (**C**) The average intensity of FM 1–43 labeling in *cib3* mutant hair cells is similar to controls. In contrast, in *cib2* mutants, hair cells that label with FM 1–43, have a significantly reduced intensity compared to controls (n=8 neuromasts per genotype in B and C). (**D–G**) Representative neuromasts labeled with FM 1–43 for each genotype. FM 1–43 label is overlaid onto laser scanning DIC images. (**H**) Schematic showing the localization of membrane-localized GCaMP6s (memGCaMP6s); the indicator used to measure Ca²⁺ influx into lateral-line hair bundles. The plane used for Ca²⁺ imaging and to determine if a hair cell is mechanosensitive is indicated by the dashed line.

*Figure 5 continued on next page*

*Figure 5 continued*

(**I-I'**) Representative example of a hair bundle imaging plane (**I**), along with the resulting Ca$^{2+}$ responses (**I'**) from hair bundles from a control neuromast. The ROIs used to measure Ca$^{2+}$ signals in each hair bundle are indicated in (**I**). (**J–L**) Representative examples of Ca$^{2+}$ responses from neuromasts in *cib2* (**J**), *cib3* (**K**), and *cib2;cib3* mutants (**L**). FM 1–43 imaging and behavior were performed at 5 dpf. Ca$^{2+}$ imaging was performed at 5 or 6 dpf. A Kruskal-Wallis test was used in A; A one-way ANOVA was used in B and C. The scale bar in D and I=5 µm.

The online version of this article includes the following figure supplement(s) for figure 5:

**Figure supplement 1.** Summary of zebrafish *cib2* and *cib3* alleles used in study.

**Figure supplement 2.** Summary of mechanosensitive calcium responses in zebrafish.

for Cib2 in MET. To assess whether Cib3 can substitute for Cib2, we examined FM 1–43 labeling in *cib2;cib3* mutants. In *cib2;cib3* mutant animals we observed a complete absence of FM 1–43 labeling in lateral-line hair cells (*Figure 5B, D and G*). Thus, taken together the results of our FM 1–43 labeling analysis are consistent with a requirement for both Cib2 and Cib3 to ensure normal MET in all lateral-line hair cells.

To obtain a more quantitative assessment of MET we used Ca$^{2+}$ imaging to perform evoked measurements of hair cell function. For this assessment, we used a transgenic line that expresses a membrane-localized GCaMP6s (memGCaMP6s) specifically in hair cells (*Figure 5H–I'*). This transgenic line has previously been used to measure Ca$^{2+}$ influx into apical hair bundles while stimulating lateral-line hair cells with a fluid-jet (*Figure 5H–I'*; *Lukasz and Kindt, 2018*). Using this approach, we were able to stimulate and monitor stimulus-evoked mechanosensitive Ca$^{2+}$ responses in hair bundles within neuromast organs of each genotype (Representative responses: *Figure 5I–L*). We quantified the number of hair bundles per neuromast with mechanosensitive Ca$^{2+}$ responses, and found that compared to controls, significantly fewer cells were mechanosensitive in *cib2* and *cib2;cib3* mutants (*Figure 5—figure supplement 2A*, control: 92.2 ± 2.5; *cib2*: 49.9 ± 5.8, *cib2;cib3*: 19.0 ± 6.6, p<0.0001). In contrast, in *cib3* mutants, the percent of mechanosensitive cells per neuromast was similar compared to controls (*Figure 5—figure supplement 2A*, *cib3*: 94.5 ± 1.8, p=1.0). We also quantified the magnitude of the mechanosensitive Ca$^{2+}$ signal in the hair bundles, focusing on only the responsive hair cells for each genotype. This quantification revealed that, while *cib3* mechanosensitive responses were not different compared to controls (*Figure 5—figure supplement 2B–C*, GCaMP6s % ΔF/F control: 70.6±5.9; *cib3*: 71.6±5.2, p>1.0), the magnitude of responses in *cib2* and *cib2;cib3* hair bundles were significantly reduced (*Figure 5—figure supplement 2B–C*, GCaMP6s % ΔF/F control: 70.6 ± 5.9; *cib2*: 33.0 ± 5.8; *cib2;cib3*: 14.7 ± 1.9, p<0.0001). Overall, our in vivo FM 1–43 and Ca$^{2+}$-imaging analyses revealed that loss of Cib2 alone, but not Cib3 can impair mechanosensory function in the majority of zebrafish lateral-line hair cells. Furthermore, Cib2 and Cib3 are both required to ensure normal MET function in all lateral-line hair cells.

## Cib2 is required in specific subsets of hair cells in the zebrafish lateral line and inner ear

Previous work has shown that within the zebrafish inner ear and lateral line there are distinct hair cell subtypes that rely on different complements of the mechanosensitive ion channel Tmc: Tmc1, 2a, and 2b. In the primary posterior lateral line (pLL) hair cells are oriented to respond to either anterior or posterior fluid flow. The subset of hair cells that detects posterior flow relies on Tmc2b while the subset that detects anterior flow relies on both Tmc2b and Tmc2a (*Figure 6A*; *Chou et al., 2017*). Furthermore, work with the crista of the zebrafish inner ear has shown that there are two morphologically distinct types of hair cells: short, teardrop-shaped cells with taller hair bundles and tall, gourd-shaped cells with shorter hair bundles (*Smith et al., 2020*; *Zhu et al., 2020*). Short cells rely on Tmc2a, while tall cells rely on either Tmc1 or Tmc2b for mechanosensitive function (*Figure 6E*; *Smith et al., 2020*). Based on this work, we hypothesized that Cib2 or Cib3 could have specific roles in zebrafish hair cell sensory organs linked to hair cell orientation in the lateral line or morphology type in the crista of the inner ear.

Our FM 1–43 labeling in neuromasts revealed no defect in *cib3* mutants, suggesting that Cib2 can compensate for Cib3 in all pLL hair cells. In contrast to *cib3* mutants, only a subset of hair cells per neuromast were labeled with FM 1–43 in *cib2* mutants (*Figure 5E*). Previous work has shown that in the pLL of *tmc2b* mutants, a subset of hair cells that respond to anterior flow label with FM 1–43 (*Chou*

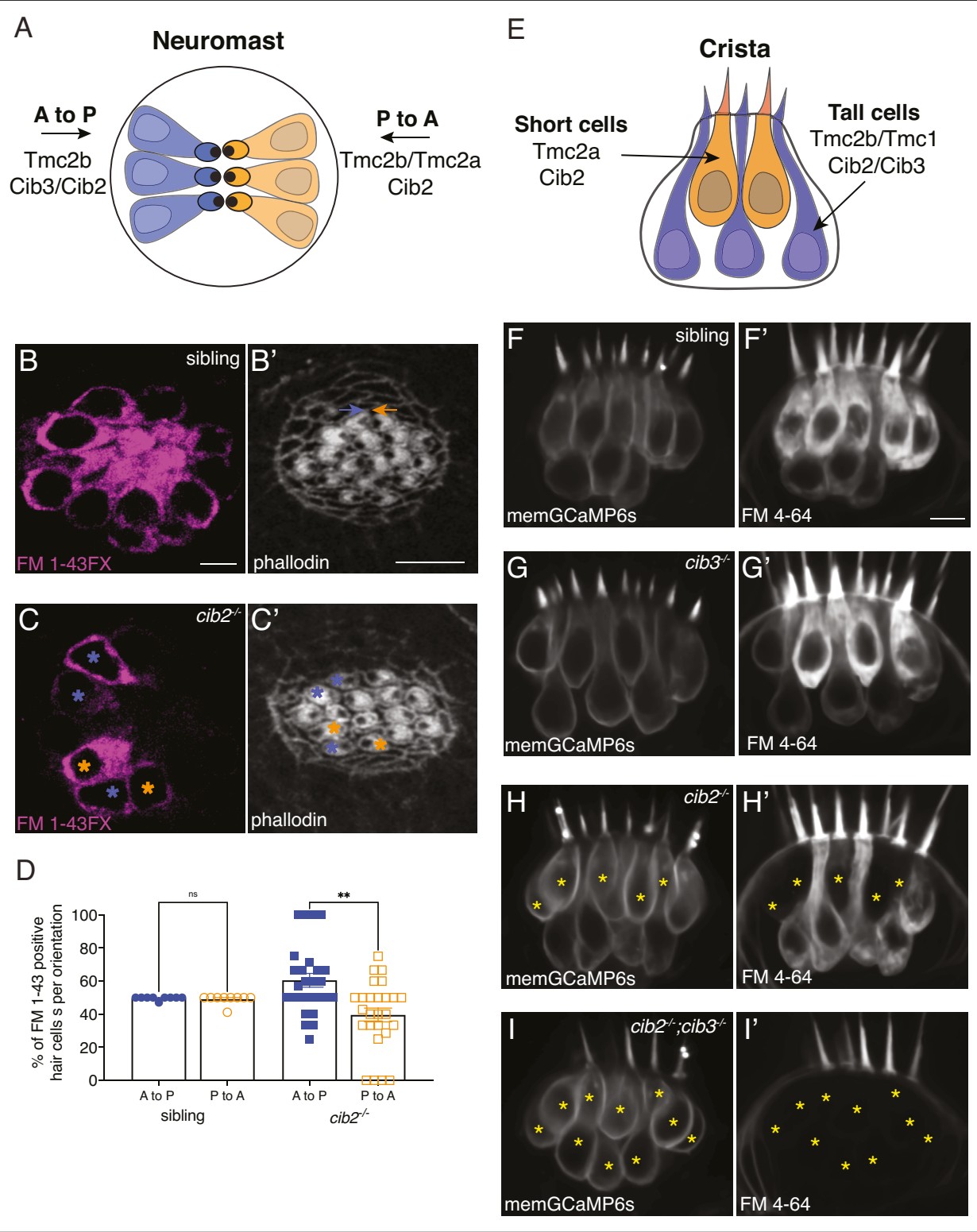

**Figure 6.** Cib2 is required in specific subsets of hair cells in the zebrafish posterior lateral line (pLL) and inner ear. (**A**) Overview of Tmc requirements in pLL hair cells. Hair cells that sense posterior flow (orange) rely on Tmc2b, while hair cells that sense anterior flow (blue) can rely on Tmc2b and/or Tmc2a for mechanosensitive function. (**B-C'**) FM 1–43 FX labeling in pLL neuromasts from sibling control (**B**) and *cib2* mutants (**C**). The residual cells in *cib2* mutants rely on Cib3 for mechanosensitive function. Phalloidin label can be used to link FM 1–43 FX label to the orientation of pLL hair cells (C,C'; see colored asterisks). In B', orange and blue arrows rest above hair cells that are oriented to respond to anterior and posterior flow, respectively. (**D**) Quantification reveals that a similar percentage of anterior and posterior responsive pLL hair cells label with FM 1–43 FX in sibling controls (posterior

*Figure 6 continued on next page*

*Figure 6 continued*

flow: 49.7%, anterior flow: 49.2%, n=9 neuromasts). In contrast, in *cib2* mutants, a significantly higher percentage of posterior responsive pLL hair cells label with FM 1–43 FX in *cib2* mutants (posterior flow: 60.5%, anterior flow: 39.5%, n=26 neuromasts). (**E**) Overview of Tmc and Cib requirements in the hair cells of zebrafish cristae. Short teardrop-shaped cells rely primarily on Tmc2a, while tall gourd-shaped cells rely primarily on Tmc2b and Tmc1. (**F-I'**) Medial cristae for each genotype. MemGCaMP6s labels all hair cells, while FM 4–64 labels hair cells with intact mechanosensitive function. In siblings and *cib3* mutants, both teardrop and gourd cells label with FM 4–64 (**F-F'**, **G-G'**). In *cib2;cib3* double mutants, no hair cells label with FM 4–64 (asterisks) (**I-I'**). In *cib2* mutants many short cells fail to label with FM 4–64 (asterisks) (**H-H'**), (n=13 double het sibling, 7 *cib3*, 14 *cib2,* and 4 double mutant crista). All images were acquired at 5 dpf. A Kruskal-Wallis test was used in D; p<0.01. The scale bars in B, B', and F'=5 µm.

*et al., 2017*). This residual FM 1–43 label is dependent on Tmc2a. Therefore, we examined whether the remaining, FM 1–43 positive hair cells in *cib2* mutants have a specific hair cell orientation. For this analysis, we used a fixable form of FM 1–43, FM 1–43 FX to label hair cells. After FM 1–43 FX labeling, we fixed the larvae and labeled the apical hair bundles with phalloidin to reveal hair cell orientation. This analysis revealed that in sibling controls, a similar percentage of FM 1–43 FX positive hair cells are oriented to respond to anterior and posterior flow (*Figure 6B, B' and D*). In contrast, when we examined *cib2* mutants, we found that significantly more of the FM 1–43 positive hair cells were oriented to respond to posterior flow (*Figure 6C–D*). This indicates that in *cib2* mutants, the residual Cib3 present in pLL hair cells preferentially functions in hair cells that express primarily Tmc2b and are oriented to respond to posterior flow (*Figure 6A*). On the other hand, Cib2 can function with hair cells that express Tmc2b (respond to posterior flow) as well as hair cells that express Tmc2b and Tmc2a (respond to anterior flow).

Next, we examined whether different hair-cell subtypes (short versus tall morphology) in the zebrafish inner ear preferentially rely on Cib2 or Cib3 for mechanosensitive function. To examine MET function, we injected FM 4–64 into the zebrafish ear in a transgenic background that labels the hair cell membrane (memGCaMP6s). FM 4–64 is a redshifted version of FM 1–43 and provides a similar readout of MET (*Erickson et al., 2017*; *Smith et al., 2020*). After injection, we imaged FM 4–64 label in the medial cristae, an inner ear sensory organ that allows for excellent delineation of short and tall cell types (*Figure 6F–F'*). We found that similar to our FM 1–43 results in the pLL, no hair cells in the medial cristae were FM 4–64 positive in *cib2;cib3* double mutants (*Figure 6I–I'*). In addition, similar to the pLL we observed no defects in FM 4–64 labeling in the crista of *cib3* mutants compared to controls (*Figure 6G–G'*). Interestingly we found that short hair cells in the crista of *cib2* mutants failed to label with FM 4–64 (*Figure 6H–H'*). Previous work has shown that short hair cells rely on Tmc2a for mechanosensitive function (*Smith et al., 2020*). Overall, our functional assessment of MET based on hair cell subtypes within the zebrafish pLL and inner ear suggests that Cib2 is required for mechanosensitive function in hair cells that express Tmc2a.

## CIB proteins simultaneously interact with at least two cytoplasmic domains of TMC1

To gain molecular insight into the role played by CIB proteins in hair-cell MET, we generated Alpha-Fold 2 (AF2) (*Jumper et al., 2021*; *Tunyasuvunakool et al., 2021*) models of the human TMC1 dimer, alone and in complex with human CIB2/3 (*Figure 7A–B*; *Figure 7—figure supplement 1*; *Figure 7—video 1*). The AF2 model of TMC1 alone resembles previous TMEM16/OSCA-based models of TMC1 and is also consistent with a recent cryo-EM structure of the worm TMC-1 protein (*Ballesteros et al., 2018*; *Jeong et al., 2022*; *Pan et al., 2018*; *Walujkar et al., 2021*). In these models, TMC1 has ten transmembrane domains ($\alpha1$ to $\alpha10$; *Figure 7—figure supplement 1*; *Figure 7—source data 1*) and a putative pore is formed by transmembrane helices $\alpha4$, $\alpha5$, $\alpha6$, and $\alpha7$ at the periphery of each monomer (*Pan et al., 2018*; *Walujkar et al., 2021*), away from the dimeric interface involving $\alpha10$. Our new models feature an additional amphipathic helix, which we denote $\alpha0$, extending almost parallel to the expected plane of the membrane bilayer without crossing towards the extracellular side (as observed for a mostly hydrophobic $\alpha0$ in OSCA channels and labeled as H3 in the worm TMC-1 structure) (*Jeong et al., 2022*; *Jojoa-Cruz et al., 2018*; *Liu et al., 2018*; *Zhang et al., 2018b*). Extracellular and intracellular domains linking transmembrane helices, which were either omitted or poorly predicted in previous work (*Ballesteros et al., 2018*; *Pan et al., 2018*; *Walujkar et al., 2021*), have well-defined secondary structure and a high-confidence score (pLDDT >70 for most regions; *Figure 7—figure supplement 1A*). The intracellular domain linking helices $\alpha2$ and $\alpha3$, denoted here as

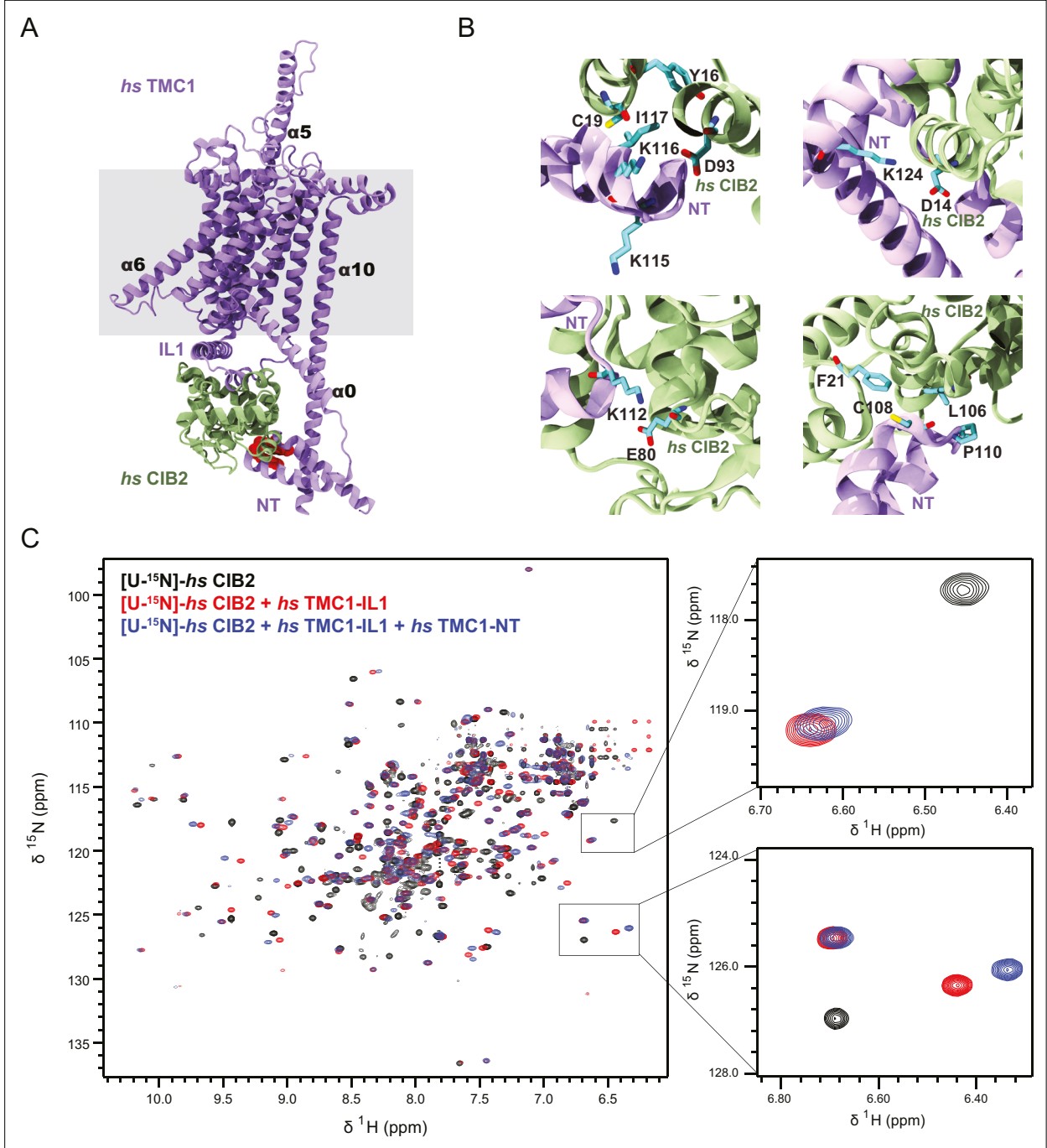

**Figure 7.** AF2 predictions and nuclear magnetic resonance (NMR) data support a clamp-like model for the human TMC1 and CIB2 complex. (**A**) AF2 model of *hs* TMC1 in complex with *hs* CIB2. Side view of monomeric *hs* TMC1 (light purple) in complex with *hs* CIB2 (light sea green). N-terminal residues of TMC1 known to interact with CIB2 are in red. (**B**) Top left panel shows residues at the *hs* TMC1-NT fragment (115-117) shown to weaken interaction with CIB2 upon mutation (*Liang et al., 2021*) at the interface with CIB2. Additional panels show details of contacts between TMC1 and CIB2 with formation of K124:D14 and K112:E80 salt bridges, as well as hydrophobic interactions. (**C**) Overlay of ¹H-¹⁵N TROSY-HSQC spectra of *hs* [U-¹⁵N]-CIB2 (black) alone and bound to either only *hs* TMC1-IL (red) or both *hs* TMC1-IL and *hs* TMC1-NT (blue). NMR data were obtained in the presence of 3 mM CaCl₂. These data are also shown in *Figure 7—figure supplement 6*.

The online version of this article includes the following video, source data, and figure supplement(s) for figure 7:

**Source data 1.** TMC sequence alignment.

**Figure supplement 1.** Overview of TMC1/2 and CIB2/3 AF2 models.

**Figure supplement 2.** CIB2 sequence alignment.

*Figure 7 continued on next page*

*Figure 7 continued*

**Figure supplement 3.** CIB3 sequence alignment.

**Figure supplement 4.** CIB sequence alignment.

**Figure supplement 5.** Size exclusion chromatography (SEC) of *hs* CIB2 and *hs* CIB3 either refolded alone or co-refolded with *hs* TMC1-IL1, and *hs* TMC1-NT.

**Figure supplement 5—source data 1.** Original SDS-PAGE gels and raw data for SEC plots.

**Figure supplement 6.** ¹H-¹⁵N TROSY-HSQC spectra of (**A**) *hs* CIB2, *hs* CIB2 + *hs* TMC1-IL1, and *hs* CIB2 + *hs* TMC1-IL1+ *hs* TMC1-NT individually (overlayed in *Figure 7C*).

**Figure 7—video 1.** Overview of *hs* TMC1 (light and dark purple) in complex with *hs* CIB2 (light sea green)-Ca²⁺ (green spheres). https://elifesciences.org/articles/89719/figures#fig7video1

IL1, adopts a helix-loop-helix with the two helices running parallel to each other and differing in length (*Figure 7—figure supplements 1–4*; *Figure 7—source data 1*). This is the same fold observed in its crystal structure in complex with CIB3 (*Liang et al., 2021*), which validated the modeling approach.

Interestingly, AF2 models of TMC1 in complex with CIB2 suggest that the N-terminus (NT) of TMC1 also interacts with CIB2 (*Figure 7A–B*). In our model, CIB2 is clamped between TMC1-NT and TMC1-IL1 while the first helix of TMC1-IL1 runs parallel to the membrane plane interacting with the base of α1 and the intracellular linkers between α4 and α5, α6 and α7, and α8 and α9. Notably, the predicted interaction between TMC1-NT and CIB2 is consistent with previous biochemical data (*Giese et al., 2017*), while the overall model resolves seemingly contradictory results indicating that either the NT or IL1 of TMC1 was responsible for interactions with CIB proteins (*Giese et al., 2017*; *Liang et al., 2021*). Additional AF2 models of TMC1 with CIB3 and of TMC2 with CIB2/3 (*Figure 7—figure supplement 1*) predicted the same architecture for the TMC1/2 + CIB2/3 complexes where CIB proteins are clamped between the IL1 and NT domains of TMCs. Predictions for fish complexes also show the same general architecture (*Figure 7—figure supplement 1A*).

Nuclear magnetic resonance (NMR) experiments were used to validate the clamp model for *Homo sapiens* (*hs*) TMC + CIB interactions. We produced uniformly ¹⁵N labeled *hs* CIB2 (22.7 kDa) and CIB3 (22.9 kDa) proteins ([U-¹⁵N]-*hs* CIB2/3), alone or co-refolded with unlabeled *hs* TMC1-IL1 (6.3 kDa). In parallel, we produced an unlabeled soluble TMC1-NT fragment (6.9 kDa) and used these CIB and TMC1 samples to test protein-protein interactions by monitoring their effects on 2D ¹H-¹⁵N-correlation spectra of the labeled CIB2 and CIB3 proteins. All protein fragments were purified using size exclusion chromatography (SEC) (*Figure 7—figure supplement 5*), revealing a single monodisperse peak for [U-¹⁵N]-*hs* CIB2 (69.2 mL elution volume in S75 16/600 column) and a slightly displaced peak for [U-¹⁵N]-*hs* CIB2 + *hs* TMC1-IL1 (68.5 mL). In contrast, SEC experiments revealed two peaks for [U-¹⁵N]-*hs* CIB3 alone (potential dimer at 62.8 mL and potential monomer at 73.1 mL) and a single peak for [U-¹⁵N]-*hs* CIB3 + *hs* TMC1-IL1 (71.3 mL). ¹H-¹⁵N-correlation spectra of [U-¹⁵N]-*hs* CIB2 alone were consistent with prior data indicating that many, but not all potential peaks are resolved (*Figure 7C*, *Figure 7—figure supplement 6A*; *Huang et al., 2012*). Spectra for the [U-¹⁵N]-*hs* CIB2 + *hs* TMC1-IL1 complex exhibit widespread perturbations, consistent with extensive remodeling of the CIB2 structure. Moreover, the addition of *hs* TMC1-NT induced additional chemical shift perturbations in many, but not all peaks, demonstrating that the TMC1-IL1 and TMC1-NT ligands bind simultaneously to [U-¹⁵N]-*hs* CIB2 (*Figure 7C*, *Figure 7—figure supplement 6B*). We observed similar results when comparing spectra for [U-¹⁵N]-*hs* CIB3 alone, co-refolded with *hs* TMC1-IL1, and upon addition of *hs* TMC1-NT (*Figure 7—figure supplement 6C–D*). A control experiment with the co-refolded mix of *hs* CIB3 + *hs* TMC1-IL1 and [U-¹⁵N]-*hs* CIB2 (*Figure 7—figure supplement 6E*) revealed no change in chemical shifts indicating that direct heteromeric interactions between CIB2 and CIB3 are unlikely. Thus, taken together, the NMR data strongly support a model in which human CIB2/3 proteins are clamped between TMC1-IL1 and TMC1-NT as suggested by AF2 and consistent with prior biochemical data revealing interactions with either TMC1-NT or TMC1-IL1 (*Giese et al., 2017*; *Liang et al., 2021*).

## Simulations reveal that CIB proteins stabilize TMC1 and TMC2

Our dimeric AF2 models of TMC1/2 alone and in complex with CIB2/3 are consistent with previous experimental data that validated the location of the putative pore in similar TMEM16/OSCA-based

homology models of TMC1 (*Pan et al., 2018*). Our models are also consistent with our NMR data supporting the clamp model for the TMC + CIB complex (*Figure 7C*). To improve the AF2 predictions and to gain mechanistic insights into TMC and CIB function we turned to all-atom equilibrium molecular dynamics (MD) simulations (*Karplus and Petsko, 1990*). We built ten different models that included dimeric TMC1 or TMC2 alone, in complex with CIB2 or CIB3 with and without bound $Ca^{2+}$. These models were embedded in either pure POPC or stereocilia-like mixed composition bilayers and solvated (150 mM KCl) to generate a total of twenty systems that were equilibrated for 100 ns each (*Figure 8A*, *Figure 8—source data 1*).

Equilibrium simulations of TMC1 and TMC2 alone revealed stable conformations for the transmembrane domains of these proteins, but a highly mobile NT domain (*Figure 8B*, *Figure 8—figure supplement 1*). In simulations where CIB2 or CIB3 were present, with or without bound $Ca^{2+}$, we observed more stable conformations of N-termini along with stable salt bridges (*Figure 8C*). Buried surface area (BSA) for the TMC + CIB complexes (*Figure 8—figure supplement 2A, C and E*, *Figure 8—source data 2*) revealed a possible trend in which TMC1 and TMC2 have more contacts with CIB3 than with CIB2. While additional sampling and longer simulations might be needed for a robust prediction, these results suggest variable affinities for TMC + CIB protein complexes that might be modulated by $Ca^{2+}$ (*Figure 8—figure supplement 2F*).

Additional analysis of the putative pore in our simulations of TMC1 and TMC2 revealed hydration as observed in previous simulations of TMEM16/OSCA-based TMC1 models, as well as movement of lipids that lined up the predicted permeation pathway for cations suggesting a membrane-based gating mechanism (*Pan et al., 2018*; *Sukharev and Corey, 2004*; *Walujkar et al., 2021*). Longer, Anton 2, 960-ns long equilibrium trajectories of TMC1 with CIB2 and bound $Ca^{2+}$ in a mixed bilayer and TMC1 with CIB2 and (un)bound $Ca^{2+}$ in a POPC bilayer revealed how parts of the amphipathic α0 helices spontaneously partitioned at the membrane interface with their hydrophobic sides pointing toward the lipid bilayer core (*Figure 8D*). The N-terminal cytoplasmic domain of LHFPL5 has been shown to interact with this TMC1 amphipathic helix, which might be essential for channel function in mice (*Qiu et al., 2023*). The BSA for the dimeric TMC1 + CIB2 complex in these longer simulations (*Figure 8—figure supplement 2E*) decreased as contacts between TMC1-α0 and CIB2 were lost for one of the monomers in the mixed bilayer system. In the monomer in which contacts were maintained, a single cholesterol was situated in the middle of the putative pore. Similarly, TMC1-α0 and CIB2 contacts were lost for both monomers in the POPC bilayer systems. Regardless, in all Anton 2 equilibrium trajectories contacts between CIB2 and the TMC1-IL1 and -NT remained, thus confirming the stability of the complex over a microsecond timescale.

## TMC1 and TMC2 complexes with CIB2 and CIB3 are predicted to be cation channels

To further explore the properties of our models of TMC + CIB complexes we took equilibrated TMC1 systems and applied a –0.5 V transmembrane voltage while monitoring ion permeation through the putative pore during 100-ns long trajectories (*Figure 8—figure supplements 3–4*, *Figure 8—source data 2*, *Figure 8—video 1*). In systems with pure POPC bilayers, we observed permeation of $K^+$ in either one or both pores of the TMC1 dimer, with or without CIB2 or CIB3 and with or without bound $Ca^{2+}$, despite the presence of $Cl^-$ (150 mM KCl). Similar results were obtained for four TMC1 systems simulated at –0.25 V. Variability of the conductance state was attributed to partial blockage by lipids and poor sampling, but in all cases in which a clear conductive state was monitored, we observed the almost exclusive passage of $K^+$ with conductance values that reached up to ~138 pS for a single pore. While simulations tend to overestimate bulk KCl conductivity and in some cases, predictions need to be scaled down (by a factor of ~0.6) (*Walujkar et al., 2021*), the conductance values monitored here are consistent with previous simulations and with the expected conductance of the inner-ear transduction channel (*Beurg et al., 2021*; *Walujkar et al., 2021*). In contrast, we observed smaller conductance values for TMC1 systems with mixed lipid bilayers (at most ~22 pS for a single pore), regardless of the presence of CIB2 or CIB3, with or without bound $Ca^{2+}$ (*Figure 8—figure supplement 4B*). Additional Anton 2 simulations for one of these TMC1 + CIB2 + $Ca^{2+}$ systems with a mixed membrane, equilibrated for 240 ns and then subjected to –0.43 V for 480 ns, revealed a low conductance value of ~6.2 pS. The reduced conductance values observed for mixed-membrane systems were attributed in part to pore blockage by cholesterol and lipids in the initial setup. Interestingly, Anton

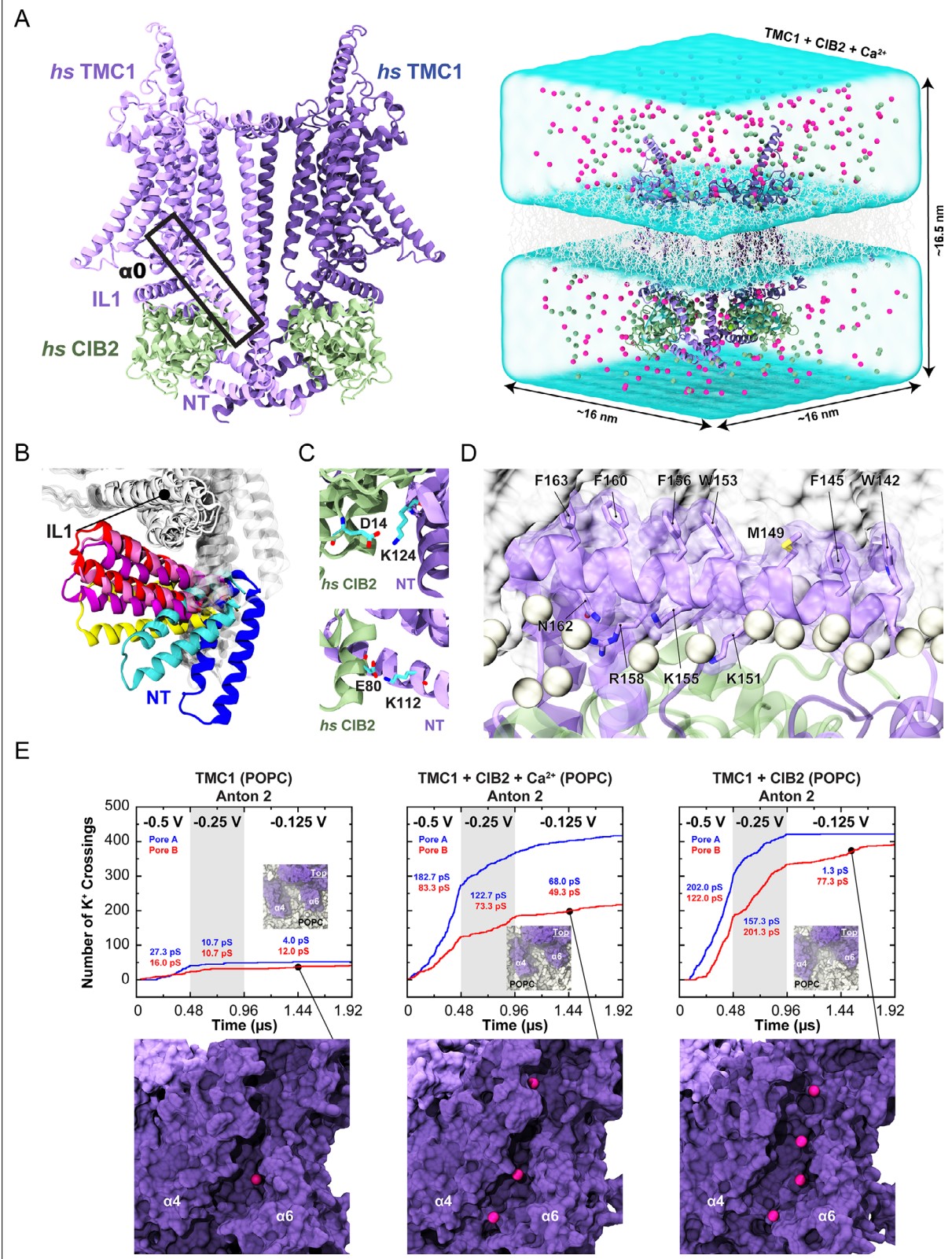

**Figure 8.** Simulations of AF2 predictions show that human TMC1 and CIB2 complexes form cation channels. (**A**) *Left:* AF2 model of the dimeric *hs* TMC1 + *hs* CIB2 complex. *Right:* A complete simulation system of *hs* TMC1 in complex with *hs* CIB2-Ca$^{2+}$ is shown in surface representation. *hs* TMC1 is shown in light (monomer A) and dark (monomer B) purple, *hs* CIB2 in light sea green, water in transparent light blue, lipid membrane in gray lines, and K$^+$ and Cl$^-$ ions are shown as pink and green spheres, respectively. (**B**) The *hs* TMC1-NT domain explores diverse conformational space in the

*Figure 8 continued on next page*

*Figure 8 continued*

absence of CIB during equilibrium simulations (S3a). (**C**) Salt bridges K124:D14 and K112:E80 remain throughout equilibrium simulations. (**D**) Snapshot of amphipathic helix α0 inserting into the lipid bilayer (S4c). (**E**) Number of $K^+$ crossings as a function of time for pores A and B of *hs* TMC1, *hs* TMC1 + *hs* CIB2 + $Ca^{2+}$, and *hs* TMC1 + *hs* CIB2 in a POPC bilayer at –0.5 V, –0.25 V, and –0.125 V. Insets show top views of pore region with TMC1 (dark purple) and lipids (light gray) in surface representation (protein and lipids partially excluded for clarity). Representative images of $K^+$ permeating the pore region between α4 and α6 of *hs* TMC1 are shown with protein in surface representation and ions as pink spheres. Conductance values calculated for $K^+$ crossings. Plots are concatenated from *Figure 8—figure supplement 4*.

The online version of this article includes the following video, source data, and figure supplement(s) for figure 8:

**Source data 1.** Summary of simulations.

**Source data 2.** Ion conduction.

**Figure supplement 1.** Stability of TMC + CIB complexes, monomers, and subdomains.

**Figure supplement 1—source data 1.** Raw data for RMSD plots.

**Figure supplement 2.** Buried surface area (BSA) in *hs* TMC1/2 and *hs* CIB2/3 complexes.

**Figure supplement 2—source data 1.** Raw data for BSA in *hs* TMC1/2 and *hs* CIB2/3 complexes.

**Figure supplement 3.** Number of $K^+$ crossings as a function of time for pores A and B of *hs* TMC1 and *hs* TMC2.

**Figure supplement 3—source data 1.** Raw data for number of $K^+$ crossings as a function of time for pores A and B of *hs* TMC1 and *hs* TMC2.

**Figure supplement 4.** Conduction events in long timescale simulations and summary of predicted conductance values.

**Figure supplement 4—source data 1.** Raw data for conduction events in long timescale simulations and summary of predicted conductance values.

**Figure supplement 5.** Possible combinations of TMC1, TMC2, CIB2, and CIB3 complexes.

**Figure 8—video 1.** Side view of the *hs* TMC1 channel pore during a trajectory at –0.5 V (monomer B).

https://elifesciences.org/articles/89719/figures#fig8video1

2 simulations of TMC1, TMC1 + CIB2, and TMC1 + CIB2 + $Ca^{2+}$ in pure POPC membranes subjected to –0.5 V, –0.25 V, and –0.125 V for 480 ns, 480 ns, and 960 ns, respectively, revealed a trend in which TMC1 exhibits higher conductance with CIB2 bound (*Figure 8E*, *Figure 8—source data 2*). This may begin to explain why CIB2 is essential for normal TMC1 function.

Simulations of TMC2 with an applied transmembrane voltage of –0.5 V revealed a more closed state when compared to our TMC1 model, regardless of membrane composition and presence or absence of CIB2, CIB3, and bound $Ca^{2+}$ (*Figure 8—figure supplement 4*). Conduction of $K^+$ was observed in most cases, but the maximum conductance did not reach above ~64 pS, suggesting that the conformation simulated is less open than the ones explored in simulations of TMC1. We also speculate that this is due to TMC2 having an intrinsic lower single-channel conductance than TMC1, as has been suggested by some experiments (*Kim et al., 2013*), but not others (*Pan et al., 2013*). It is also possible that our TMC2 model is not in a fully open conformation, which can only be reached upon mechanical stimulation. Regardless, our simulations unequivocally indicate that TMC1 and TMC2 are cation channels and that these proteins can form stable complexes with CIB2 and CIB3.

## Discussion

Previous work on CIB proteins had primarily focused on mammalian auditory hair cells, leaving open the question of whether they play a functional role in vestibular function and across vertebrate species. Recently, *Wang et al., 2023* have suggested that CIB2 and CIB3 are important for stereocilia maintenance and MET in mouse vestibular hair cells. This work demonstrated that similar to *Tmc1* and *Tmc2*, *Cib2* is highly expressed in the striolar region while *Cib3* is expressed in the extrastriolar regions, and mice lacking both *Cib2* and *Cib3* had no MET and deficits in vestibular hair cell bundle structure (*Wang et al., 2023*). Taking a step forward, here we establish that CIB2 and CIB3 are highly conserved regulators of MET in both mouse vestibular hair cells and zebrafish hair cells. We demonstrate that vestibular hair cells in mice and zebrafish lacking CIB2 and CIB3 are not degenerated but have no detectable MET, assessed via FM 1–43 dye uptake, at time points when MET function is well developed in wild-type hair cells. Similar structural-function phenotype has been reported in auditory hair cells in *Tmc1/Tmc2* double mutant mice (*Kawashima et al., 2015*). Besides the known in vitro interactions (*Giese et al., 2017*; *Liang et al., 2021*), our study provides strong genetic evidence for

the direct functional relationship between CIB and TMC proteins in regulating MET both in auditory and vestibular hair cells.

Previous studies have shown that the loss of CIB2 leads to profound hearing loss from birth in both humans and mice (*Giese et al., 2017*; *Liang et al., 2021*; *Seco et al., 2016*). This suggests that the endogenous levels of CIB3 present in auditory hair cells are not sufficient to compensate for CIB2 in hearing. Prior work has also shown that over-expression of CIB3 in mouse auditory outer hair cells can only partially compensate for the loss of CIB2 in MET (*Liang et al., 2021*). However, whether overexpression of CIB3 can rescue impaired hearing thresholds in *Cib2* mutant mice is not known. While CIB2 is also present in vestibular hair cells, compared to auditory hair cells, the expression of CIB3 is several folds higher in vestibular end organs. Indeed, our current data demonstrate that, unlike auditory hair cells, endogenous levels of CIB3 can fully compensate for the loss of CIB2 in vestibular hair cells in mice. Together our work on the vestibular system in mice highlights that both CIB2 and CIB3 can play overlapping roles in MET function.

Interestingly, our corresponding work in zebrafish indicates that Cib3 can also compensate for Cib2, but only in specific subsets of hair cells. Recent genetic, morphology, and scRNAseq data have revealed that within the zebrafish inner ear, each sensory epithelia (utricle, saccule, cristae) has distinct groups of hair cells (*Shi et al., 2023*; *Smith et al., 2020*). In the crista of the zebrafish inner ear, which responds to angular acceleration, there are two main groups of hair cells that can be delineated by distinct morphological and genetic features. Morphologically these two groups are defined as either short or tall hair cells (*Figure 6A and E*). Genetic studies have revealed that these two types of hair cells rely on different Tmc combinations for MET–short cells rely on Tmc2a while tall cells rely on Tmc2b and Tmc1 (*Smith et al., 2020*). More recent mRNA in situ studies have validated this *Tmc* requirement and shown that *tmc2a, tmc2b, tmc1* are expressed in just tall cells, while *tmc2a* is only expressed in short cells (*Smith et al., 2023*). Interestingly, this work also demonstrated that *cib2* is expressed broadly in tall and short hair cells, while *cib3* showed more restricted expression in tall hair cells. Our work on MET in the zebrafish cristae indicates that Cib2 is required in both short and tall cells, while Cib3 is required in just tall cells. This expression data set is supported by the absence of FM 4–64 labeling from short hair cells when Cib2 is lost (*Figure 6H*). In contrast, the loss of Cib3 alone does not impact FM 4–64 labeling in the zebrafish crista, indicating that Cib2 can compensate for the loss of Cib3 in both short and tall cells (*Figure 6G*). Currently, it is not clear what role different types of hair cells play in zebrafish inner ear epithelia. Our present findings show that zebrafish *cib2* mutants have a severely diminished acoustic startle response, a behavior mediated primarily by the saccule (*Figure 5A*). These behavioral results suggest that Cib2 plays an important role in hair cells that detect acoustic stimuli.

In addition to the inner ear, we likewise found that both Cib2 and Cib3 play an important role in the zebrafish lateral line. Similar to the zebrafish inner ear and mouse vestibular system, we show that in lateral-line hair cells, Cib2 can compensate for MET in all hair cells when Cib3 is lost (*Figure 5B,C,F,K*, *Figure 5—figure supplement 2A-C*). However, like in the zebrafish inner ear, when Cib2 is lost, MET is lost from a specific population of lateral-line hair cells. In the lateral line, each neuromast has two cell populations that detect flow in opposing directions (*López-Schier and Hudspeth, 2006*; *Pujol-Martí and López-Schier, 2013*). Work in the posterior lateral line has demonstrated that cells that detect anterior flow rely on Tmc2a/2b for MET, while cells that detect posterior flow rely on just Tmc2b (*Chou et al., 2017*). In our work, we find that cells that detect anterior flow primarily rely on Cib2 for MET (*Figure 6C–D*). Furthermore, this work indicates that Cib3 cannot compensate for the loss of Cib2 in these cells. Interestingly, our functional assessment in the zebrafish lateral line and crista suggests that Cib2 is specifically required for mechanosensitive function in populations of hair cells that express Tmc2a. Whether Cib3 is unable to pair with Tmc2a or is simply not expressed along with Tmc2a remains an unanswered question. The significance of this pairing is also unclear. Recent work has shown that the addition of Tmc2a to lateral-line cells that detect anterior flow significantly augments the mechanosensitive responses of these cells (*Kindig et al., 2023*). Whether Cib2 is required along with Tmc2a for these augmented responses remains unclear.

Our in vivo studies of mouse TMC + CIB and zebrafish Tmc + Cib complexes suggest a conserved functional relationship between these two protein families across vertebrates. Structural predictions using AF2 show conserved folds for human and zebrafish proteins, as well as conserved architecture for their protein complexes. Predictions are consistent with previous experimentally validated models

for the TMC1 pore (*Ballesteros et al., 2018*; *Pan et al., 2018*), with the structure of human CIB3 coupled to mouse TMC1-IL1 (*Liang et al., 2021*), and with our NMR data validating the interaction between human TMC1 and CIB2/3 proteins. Remarkably, the AF2 models are also consistent with the architecture of the nematode TMC-1 and CALM-1 complex (*Jeong et al., 2022*), despite low sequence identity (36% between human TMC1 and worm TMC-1 and 51% between human CIB2 and worm CALM-1). This suggests that the TMC + CIB functional relationship extends beyond vertebrates.

Taken together, the modeling predictions, NMR experiments, and simulations presented here provide a structural framework to interpret and understand seemingly contradictory results. In a previous study (*Giese et al., 2017*), we showed that the N-termini of TMC1 and TMC2 interact with CIB2, and subsequent work (*Liang et al., 2021*) presented biochemical data validating the interaction between the TMC1-NT with CIB2. However, X-ray crystallography data demonstrated that the main interaction between TMC1 and CIB3 involved IL1, leaving the role played by CIB2 + TMC-NT interactions unaddressed (*Liang et al., 2021*). Here, we show using NMR that both TMC1-NT and TMC1-IL1 bind to CIB2 and CIB3 non-competitively. These interactions are consistent with AF2 models for TMC + CIB complexes where CIB proteins are clamped between TMC1-NT and TMC1-IL1. Simulations of these models indicate that there is some potential preferential binding of TMC1 and TMC2 to CIB3 over CIB2 (predicted from BSA) and that TMC + CIB interactions are stable and last for microseconds, with biochemical and NMR experiments showing that these interactions are stable at even longer timescales.

Interestingly, the behavior of CIB2 and CIB3 in solution (SEC experiments using 3 mM CaCl$_2$) is different in the absence of TMC1-IL1. While CIB2 appears as a single peak, CIB3 shows two clear peaks indicating the presence of monomeric and dimeric states, consistent with the crystal structure of an apparent CIB3 dimer (*Liang et al., 2021*). In the presence of TMC1-IL1, the CIB3 dimeric peak is absent, and the potentially monomeric peak is slightly shifted accounting for the added TMC1-IL1, as observed for CIB2. This result suggests that CIB2, both in the presence and in the absence of TMC1-IL1, is monomeric in solution (*Figure 7—figure supplement 5*). This proposal is consistent with prior conclusions that CIB2 is a monomer (*Dal Cortivo et al., 2019*), but is in disagreement with other reports indicating that CIB2 is a dimer in solution (*Vallone et al., 2018*) and our own prior pull-down assays indicating that CIB2 co-immunoprecipitated with differentially tagged CIB2 or CIB3 (*Giese et al., 2017*; *Riazuddin et al., 2012*). Moreover, our NMR data (obtained using 3 mM CaCl$_2$) indicates that CIB2 + TMC1-IL1 is unlikely to directly interact with CIB3. The simplest explanation for these seemingly contradictory results comes from the TMC1 and TMC2 dimeric states (*Pan et al., 2018*) and our modeling indicating that each TMC monomer interacts with one CIB protein (*Figure 7—figure supplement 1*). Pull-down assays might simply be reflecting the oligomeric state of TMC proteins. In such a scenario, and considering that TMC proteins have been suggested to form hetero-oligomers (*Pan et al., 2018*), there are up to ten different combinations of TMC1/2 and CIB2/3 proteins (*Figure 8—figure supplement 5*) that might have distinct properties. Such variety of complexes with potentially different conductance values might explain tonotopic changes of conductance along the cochlea, although other explanations exist, including a variable number of TMC proteins around the tip link (*Beurg et al., 2018*) and the direct influence of membrane composition on conductance (*Effertz et al., 2017*; *Walujkar et al., 2021*).

How TMC + CIB interactions depend on Ca$^{2+}$ concentration may have important functional implications for adaptation and hair cell mechanotransduction. Structures of CIB3 and worm CALM-1, a CIB2 homologue, both bind divalent ions via EF-hand motifs proximal to their C-termini (*Jeong et al., 2022*; *Liang et al., 2021*). Reports on CIB2 affinities for Ca$^{2+}$ are inconsistent, with $K_D$ values that range from 14 μM to 0.5 mM (*Blazejczyk et al., 2009*; *Vallone et al., 2018*). Although qualitative pull-down assays done in the presence or the absence of 5 mM CaCl$_2$ suggest that the TMC1 and CIB2 interactions are Ca$^{2+}$-independent (*Liang et al., 2021*), strength and details of the CIB + TMC-IL1 and CIB + TMC-NT contacts might be Ca$^{2+}$-dependent, especially considering that Ca$^{2+}$ induces changes that lead to exposure of hydrophobic residues involved in binding (*Blazejczyk et al., 2009*).

Our simulations in the presence of a biasing potential unequivocally demonstrate that both TMC1 and TMC2, in the presence or the absence of CIB proteins, with or without bound Ca$^{2+}$, and in two types of membranes, are cation channels with predicted single-channel conductance values that can reach values that are consistent with those of the transduction channel. Our models emphasize again the role played by the membrane in lining the pore of TMC proteins (*Pan et al., 2018*; *Walujkar et al.,*

2021), and also indicate that CIB proteins are ideally positioned to connect various parts of cytosolic inter-helix linkers with the helices that form the pore, especially α3 and α4, recently implicated in gating (*Akyuz et al., 2022*; *Walujkar et al., 2021*). Last, our modeling also positions the myristoylated N-terminus of CIB2 and CIB3 close enough to the membrane bilayer, which might be yet another way to sense membrane stretch and influence gating. Further studies should help provide a comprehensive view into CIB function in channel assembly, activation, and potentially hair-cell adaption.

Overall, the biochemically supported atomic-resolution structural models and simulations presented here along with our results obtained using animal models strongly suggest that TMC1/2 proteins must function with CIB2/3 in normal hair-cell MET across sensory organs, hair-cell types, and species. This is supported by structural data indicating multiple contact points between TMC and CIB proteins, simulation data predicting that CIB proteins stabilize TMC cytosolic domains and may enhance cation conduction, and in vitro and in vivo functional data indicating that both TMC1/2 and CIB2/3 proteins are needed for hair-cell MET.

# Materials and methods

**Key resources table**

| Reagent type (species) or resource | Designation | Source or reference | Identifiers | Additional information |
|---|---|---|---|---|
| Antibody | Goat anti-Mouse polyclonal IgG (H+L) Highly Cross-Adsorbed Secondary Antibody, Alexa Fluor 568 | Thermo Fisher Scientific | Cat#A-11031; RRID:AB_144696 | Dilution, 1:1000 |
| Antibody | Goat anti-Mouse polyclonal IgG (H+L) Secondary Antibody, Alexa Fluor 488 | Thermo Fisher Scientific | A32723 | Dilution, 1:1000 |
| Antibody | Goat anti-Rabbit polyclonal IgG (H+L) Highly Cross-Adsorbed Secondary Antibody, Alexa Fluor 488 | Thermo Fisher Scientific | Cat#A-11034; RRID:AB_2576217 | Dilution, 1:1000 |
| Antibody | Goat anti-rabbit polyclonal IgG (H+L) Secondary Antibody, Alexa Fluor 568 | Thermo Fisher Scientific | A-11011 | Dilution, 1:1000 |
| Antibody | Mouse monoclonal anti-FLAG M2 | Sigma-Aldrich | Cat#F3165; RRID:AB_25959 | Dilution, 1:1000 |
| Antibody | Mouse monoclonal anti-FLAG M2-HRP | Thermo Fisher Scientific | Cat#A8592; RRID:AB_439702 | Dilution, 1:200 |
| Antibody | Mouse monoclonal anti-V5 | Thermo Fisher Scientific | Cat# R960-25; RRID:AB_2556564 | Dilution, 1:200 |
| Antibody | Mouse monoclonal anti-V5-HRP | Thermo Fisher Scientific | Cat#R961-25; RRID:AB_2556565 | Dilution, 1:5000 |
| Antibody | Mouse monoclonal GAPDH | Santa Cruz | sc-32233 | Dilution, 1:1000 |
| Antibody | Rabbit polyclonal anti-CIB3 | Sigma | SAB2103524 | 1 µg/mL |
| Antibody | Rabbit polyclonal anti-GFP | Invitrogen | A-11122 | Dilution, 1:1000 |
| Antibody | Rabbit polyclonal anti-GFP-HRP | Thermo Fisher Scientific | Cat#A10260; RRID:AB_2534022 | Dilution, 1:500 |
| Antibody | Rabbit polyclonal anti-Myosin VIIa | Proteus Biosciences | #25–6790 | Dilution, 1:500 and 1:1000 |
| Antibody | Rabbit polyclonal anti-V5 | Abcam | Cat#ab9116; RRID:AB_307024 | Dilution, 1:1000 |
| Chemical compound, drug | FM 1–43 FX | Thermo Fisher Scientific | F35355 | |
| Chemical compound, drug | FM 1–43 | Thermo Fisher Scientific | T3163 | |
| Chemical compound, drug | FM 4–64 | Thermo Fisher Scientific | T13320 | |
| Chemical compound, drug | Alexa Fluor 488 Phalloidin | Thermo Fisher Scientific | A12379 | |

*Continued on next page*

*Continued*

| Reagent type (species) or resource | Designation | Source or reference | Identifiers | Additional information |
|---|---|---|---|---|
| Chemical compound, drug | Alexa Fluor 633 Phalloidin | Thermo Fisher Scientific | A22284 | |
| Chemical compound, drug | 16% Paraformaldehyde | Electron Microscopy Sciences | 50-980-487 | |
| Chemical compound, drug | Triton X-100 | Polysciences | 04605–250 | |
| Chemical compound, drug | DAPI (4',6-Diamidino-2-Phenylindole, Dihydrochloride) | Invitrogen | D1306 | |
| Chemical compound, drug | Protease inhibitor mixture | Roche | 11697498001 | |
| Chemical compound, drug | Protein A–Sepharose CL-4 | Thermo Fisher Scientific | 101042 | |
| Chemical compound, drug | Polyethylenimine | Polysciences | 23966 | |
| Commercial assay or kit | 4–20% Tris-Glycine gel | Novex | XV04205PK20 | |
| Commercial assay or kit | SMARTScribe Reverse Transcriptase kit | Clontech | 639537 | |
| Commercial assay or kit | SYBRgreen technology | Qiagen | 330503 | |
| Commercial assay or kit | Amersham ECL Prime Western Blotting Detection Reagent | GE Healthcare | RPN2236 | |
| Commercial assay or kit | QIAshredder | Qiagen | Cat#79656 | |
| Commercial assay or kit | Trans-Blot Turbo Transfer System | Bio-Rad | Cat#1704156 | |
| Commercial assay or kit | Trans-Blot Turbo Midi PVDF Transfer Packs | Bio-Rad | Cat#1704157 | |
| Commercial assay or kit | 4–20% Mini-PROTEAN TGX Stain-Free Protein Gels | Bio-Rad | Cat#4568094 | |
| Commercial assay or kit | 4–20% Criterion TGX Stain-Free Protein Gel | Bio-Rad | Cat#5678094 | |
| Commercial assay or kit | GFP-Trap Magnetic Agarose | Chromotek | Cat#gtma-20 | |
| Commercial assay or kit | Anti-v5 agarose affinity gel | Sigma-Aldrich | Cat#A7345 | |
| Commercial assay or kit | Anti-FLAG M2 Magnetic Beads | Sigma-Aldrich | Cat#M8823 | |
| Commercial assay or kit | Nunc Lab-Tek chambered coverglass | Thermo Fisher Scientific | Cat#155411PK | |
| Commercial assay or kit | Lipofectamine 3000 Transfection Reagent | Thermo Fisher Scientific | Cat#L3000015 | |
| Strain, strain background (mouse) | *Cib2tm1a* | EUCOMM | | |
| Strain, strain background (mouse) | *Cib3KO* | This study | N/A | |
| Strain, strain background (zebrafish) | *cib2idc21* zebrafish | This study | ZDB-ALT-221219–2 | |
| Strain, strain background (zebrafish) | *cib3idc22* zebrafish | This study | ZDB-ALT-221219–3 | |
| Strain, strain background (zebrafish) | Tg(myo6b:GCaMP6s-CAAX)*idc1* zebrafish | *Sheets et al., 2017* | ZDB-ALT-170113–3 | |

## Mouse strains and husbandry

### *Cib2*^KO mice

*Cib2* ^tm1a(EUCOMM)Wtsi^ mice were obtained from the EUCOMM repository and maintained on C57BL/6 N background. A gene trap cassette containing lacZ and neomycin resistance genes flanked by FRT sites was inserted downstream of exon 3, with exon 4 flanked by loxP sites.

## Cib3<sup>KO</sup> mice

*Cib3* knockout mice were generated using CRISPR/Cas9 technology. An indel in exon 4 of *Cib3* was introduced leading to a nonsense mutation and a premature stop at the protein level (p.Asp54*). Primers used to genotype Cib3 mutant mice were F: GTAGGGGCATGGATAACATCTG and R: GTTG AATGCTTGTCTCCCCAGG. *Cib3*<sup>KO</sup> mice were backcrossed using C57Bl6/JN *Cib2*<sup>KO</sup> mice for several generations. *Cib3* allele was validated at the genomic, RNA messenger, and protein levels.

Animals were group-housed with their littermates on a 12:12 hr light: dark cycle at 20 °C with ad libitum access to food and water. All animal procedures were approved by the Institutional Animal Care and Use Committees (IACUCs) at the University of Maryland (protocol #0420002) and at Harvard Medical School (protocol #00001240).

## Western blot and qPCR in mice

Mouse heart tissues were homogenized in CHAPS buffer containing 1 X protease inhibitors, 1 mM Na orthovanadate, 10 mM Na glycerophosphate, and 10 mM NaF. Equivalent protein concentrations were fractionated on a 4–20% Bis-Tris gel (Invitrogen) and transferred to polyvinylidene fluoride (PVDF) membrane (Millipore). Membranes were blocked and then probed with the rabbit anti-CIB2 (1:500), CIB3 (1:500), and GAPDH (1:500) overnight at 4 °C, followed by three washes of 30 min each in 1 X TBST. After washing, blots were incubated with horseradish peroxidase-conjugated anti-rabbit antibody (1:1000, Sigma NA934V, Lot # 17640116) for 2 hr at room temperature, followed by detection using the ECL Prime Western Blotting System (Thermo Fisher Scientific 32,106).

RNAs from brain were extracted from the *Cib2;Cib3* mutant and control mice using the Ribopure kit (Ambion). SMARTScribe Reverse Transcriptase kit (Clontech) was used to generate cDNAs, and SYBRgreen technology (Qiagen) was used to perform the qPCR. The following primers were used to amplify *Cib2* and *Cib3*: Cib2_exon4_F: CTCTGTGCTCTGCGAATCAG, Cib2_exon5_R: GGCCAGCG TCATCTCTAAGT, Cib3_exon3_F: TGGTGCCTCTTGACTACACG, Cib3_exon5_R: CAGCACCTTCTC ACAGACCA. *hprt* amplification was used to normalize the samples: hprt_exon8_F: TGTTGTTGGATA TGCCCTTG, hprt_exon9_R: GGCTTTGTATTTGGCTTTTCC.

## Quantification of VOR and OKR in mice

Surgical techniques and experimental setup have been previously described (*Beraneck and Cullen, 2007*). Eye movement data were collected using an infrared video system (ETL-200, ISCAN system). The rotational velocity of the turntable (head velocity) was measured using a MEMS sensor (MPU-9250, SparkFun Electronics). Eye movements during OKR were evoked by sinusoidal rotations of a visual surround (vertical black and white stripes, visual angle width of 5°) placed around the turntable at frequencies 0.2, 0.4, 0.8, 1, and 2 Hz with peak velocities of ±16°/s. To record VOR responses, the turntable was rotated at sinusoidal frequencies 0.2, 0.4, 0.8, 1, and 2 Hz with peak velocities of ±16°/s in both light and dark. In the light condition, the visual surround remained stationary, whereas in the dark condition, both the visual surround and turntable rotated in phase. Head and eye movement signals were low-passed filtered at 125 Hz and sampled at 1 kHz. Eye position data were differentiated to obtain velocity traces. Cycles of data with quick phases were excluded from the analysis. Least-square optimization determined the VOR and OKN gains and phases (*Beraneck et al., 2008*) plotted as mean ± standard deviation (SD) against all frequencies for all mice. *Cib2<sup>+/+</sup>; Cib3<sup>KO/KO</sup>* (n=4), *Cib2<sup>KO/+</sup>;Cib3<sup>KO/KO</sup>* (n=5), and *Cib2<sup>KO/KO</sup>;Cib3<sup>KO/KO</sup>* (n=4) mice were tested for all frequencies. Two-way ANOVAs and Bonferroni post hoc tests were used to test statistical significance.

## Mouse rotarod test

*Cib2<sup>+/+</sup>;Cib3<sup>KO/KO</sup>* (n=6), *Cib2<sup>KO/+</sup>;Cib3<sup>KO/KO</sup>* (n=6), and *Cib2<sup>KO/KO</sup>;Cib3<sup>KO/KO</sup>* (n=6) mice were used. Motorized rotating rod (Rotarod, IITC Life Science) was programmed to accelerate from 4 rpm to 40 rpm with a ramp of 300 s for each test trial. The time taken for a mouse to fall off the rotarod was recorded. Three trials were performed each day for five consecutive days. First day (day 0) was considered the training day.

## Quantification of head motion in mice

Head motion was recorded in mice via a miniature motion sensor (MPU-9250, TDK Corporation) attached to the top of the skull. The motion sensor recorded 3-axis linear acceleration (i.e. fore-aft,

lateral, and vertical) and 3-axis angular velocity (roll, pitch, and yaw) at a 100 Hz sampling rate. The power spectra of the motion signals in all six dimensions were computed using Welch's averaged periodogram with nfft=512 and a Bartlett window (pwelch function, MATLAB, MathWorks).

## Circling behavior in mice

Mice were placed in a cylindrical-shape container (27 cm in diameter, 30 cm in height) and their behavior was recorded using a cell-phone camera. The number of rotations made during the 120 second recording were counted offline by watching the recorded videos. A rotation is defined as a completed 360 degree rotation of the mouse's heading.

## Immunostaining in mice

The cochlear and vestibular sensory epithelia were isolated, fine dissected, and permeabilized in 0.25% Triton X-100 for 1 hr and blocked with 10% normal goat serum in PBS for 1 hr. The tissue samples were probed with primary antibody overnight and after three washes were incubated with the secondary antibody for 45 min at room temperature. Rhodamine phalloidin or Alexa fluor phalloidin 488 were used at a 1:250 dilution for F-actin labeling. Nuclei were stained with DAPI (Molecular Probes). Images were acquired using either LSM 700 laser scanning confocal microscope (Zeiss, Germany) with a 63×1.4 NA or 100×1.4 NA oil immersion objectives or Leica SP8 laser scanning confocal microscope with a 100×1.44 NA objective lens. Stacks of confocal images were acquired with a Z step of 0.05–0.5 μm and processed using ImageJ software (National Institutes of Health). Experiments were repeated at least 3 times, using at least three different animals.

## FM 1-43 dye uptake experiments in mice

In the Ahmed lab, cochlear explants from *Cib2* and *Cib3* mutant mice were dissected at postnatal day 5 (P5) and cultured in a glass-bottom petri dish (MatTek, Ashland, MA). They were maintained in Dulbecco's modified Eagle medium (DMEM) supplemented with 10% FBS (Life Technologies) for 2 d at 37 °C and 5% $CO_2$. Explants were incubated for 10 s with 3 μM FM 1–43, washed three times with Hank's balanced salt solution, and imaged live using a Zeiss LSM 700 scanning confocal microscope.

In the Holt Lab, P60 mice were injected intraperitoneally with 5 mg/g mouse weight of FM 1–43 FX fixable form as previously described (*Meyers et al., 2003*). Tissue was collected after ~24 hr and fixed for 1 hr at 25 °C in 4%PFA/1 X PBS protected from light. Tissue was then transferred into 120 mM EDTA pH 7.4 for 48–72 hr protected from light. Then tissue was transferred to 1XPBS, dissected, incubated in 1:1000 dilution of Phalloidin (Invitrogen), washed three times in 1XPBS and mounted under glass coverslips for microscopy. Imaging was completed using a 20x objective on a Zeiss 800 confocal microscope with laser intensity, detector gain, and post processing set equally for each sample.

## ABR measurements in mice

Hearing thresholds of *Cib2* and *Cib3* mutant mice (n = 5–10 for each genotype) were evaluated by recording ABRs. All ABR recordings, including broadband clicks and tone-burst stimuli at three frequencies (8, 16, and 32 kHz), were performed using an auditory-evoked potential RZ6-based auditory workstation (Tucker-Davis Technologies) with high frequency transducer RA4PA Medusa PreAmps. Maximum sound intensity tested was 100 dB SPL. TDT system III hardware and BioSigRZ software (Tucker Davis Technology) were used for stimulus presentation and response averaging.

## Zebrafish strains and husbandry

Zebrafish (*Danio rerio*) were grown at 28 °C using standard methods. Zebrafish were maintained in a Tubingen or TAB background. Larvae were raised in E3 embryo medium (5 mM NaCl, 0.17 mM KCl, 0.33 mM $CaCl_2$, and 0.33 mM $MgSO_4$, buffered in HEPES, pH 7.2). Zebrafish work at the National Institute of Health (NIH) was approved by the Animal Use Committee at the NIH under animal study protocol #1362–13. All experiments were performed on larvae at 5–6 d post fertilization (dpf). At this age, sex determination is not possible. Neuromasts (L1-L5) in the primary posterior lateral line were used for our analyses. For $Ca^{2+}$ imaging the following transgenic line was used: *Tg(myo6b:memG-CaMP6s)*[idc1] (*Jiang et al., 2017*).

*Cib2*[idc21] and *cib3*[idc22] mutants were generated in-house using CRISPR-Cas9 technology as previously described (*Varshney et al., 2016*). Exon 3 and exon 6 were targeted for *cib2* and *cib3,* respectively.

Guides for *cib2* and *cib3* are as follows: 5'-GGAGGAGCTTACACCAG(AGG)–3' and 5'-GGAAATCC TCCAGAGAAAGG(CGG)–3'. Founder fish were identified using fragment analysis of fluorescent PCR products (*Varshney et al., 2016*). Founder fish containing a 10 bp insertion in *cib2* 5'-GCTT ACACCA(CCTTTGGTAG)GAGGAGGTTA-3' resulting in a stop codon in the third EF-hand domain, at amino acid 148, was selected for analysis (*Figure 5—figure supplement 1*). In addition, a founder fish containing a 4 bp deletion in *cib3* 5'-TGGCCGCCT(TTCT)CTGGAGG-3' resulting in a stop codon in the third EF-hand domain, at amino acid 159, was selected for analysis (*Figure 5—figure supplement 1*). After behavior or imaging analyses, genotypes were confirmed using standard PCR and sequencing. The following primers were used for genotyping: *cib2*_FWD 5'- ACTGAAGCAACTCCTTTTGTGG-3' and *cib2*_REV 5'-GAACCATGCATTTTTACAGC-3' and *cib3*_FWD 5'-ATAGTCTCTCCATCTATAGA CT-3' and *cib3*_REV 5'-TTTTAGGGTGAAATAAAACAGA-3'. Crosses of *cib2*$^{+/-}$;*cib3*$^{-/-}$ X *cib2*$^{+/-}$;*cib3*$^{+/-}$ or *cib2*$^{-/-}$;*cib3*$^{+/-}$ X *cib2*$^{+/-}$;*cib3*$^{+/-}$ +/- were used to propagate larvae for our analyses. All controls or siblings are double heterozygous animals. Larvae were chosen at random, except for *cib2;cib3* homozygous mutants which, similar to other zebrafish mutants with hearing and balance defects, had no swim bladder and no acoustic startle response.

## Labeling zebrafish hair cells with FM 1-43, FM 4-64, and FM 1-43FX

To label lateral-line hair cells with FM 1–43, larvae were immersed in 1 µM FM 1–43 (Thermo Fisher, T3163) in E3 for 30 s, and then washed 3 times in E3. After washing, larvae were mounted in 1% low melt agarose containing an anesthetic, 0.02% tricaine methanesulfonate (Syndel, ANADA 200–226). FM 1–43 labeled hair cells were then imaged on an A1 Nikon upright laser-scanning confocal micro-scope with a 60×1.0 NA water objective. FIJI was used to process image Z-stacks. To estimate the number of hair cells per neuromast, laser scanning DIC images were used to count the number of hair bundles per neuromast. To quantify FM 1–43 label, a 5 µm circular region of interest (ROI) was placed on the soma of each hair cell with a hair bundle in the DIC image. These ROIs were used to quantify the mean FM 1–43 intensity per hair cell. These intensity values were averaged to calculate the average FM 1–43 intensity per neuromast.

FM 4–64 (Thermo Fisher, T13320) was used to label hair cells in the crista within the zebrafish inner ear. For this labeling, *Tg(myo6b:memGCaMP6s)$^{idc1}$* larvae were mounted on their side on Sylgard chamber in E3 embryo media containing 0.02% MESAB. Approximately 2–3 nL of 10 µM FM 4–64 in 0.1 M KCl were injected into the otic capsule as previously described (*Smith et al., 2020*). FM 4–64 labeled hair cells of the medial crista were then imaged on an A1 Nikon upright laser-scanning confocal microscope with a 25×1.10 NA or 60×1.0 NA water objective. FIJI was used to process image Z-stacks. To assess tall versus short hair cells in each crista *memGCaMP6s* label was used.

To correlate hair bundle orientation in lateral line hair cells with intact mechanotransduction, larvae were immersed in 3 µM FM 1–43 FX (Thermo Fisher, F25255) in E3 for 30 s, and then washed three times in E3. After washing, larvae were fixed with 4% paraformaldehyde in PBS at 4 °C overnight. After fixation, larvae were washed 4×5 min in 0.1% Tween in PBS (PBST). Larvae were then placed in blocking solution (2% goat serum, 1% bovine serum albumin, 2% fish skin gelatin in PBS +0.3% triton) for 2 hr at room temperature. After block Alexa Fluor 633 Phalloidin (Thermo Fisher, A22284) was added at 1:100 in PBST. Larvae were incubated for 4 hr at room temperature and then washed 4×5 min in PBST. Larvae were mounted on glass slides with Prolong Gold (Thermo Fisher Scientific) using No. 1.5 coverslips. To image hair bundle orientation and FM 1–43 FX label in lateral-line hair cells, a Zeiss LSM780 confocal microscope with Airyscan was used. Hair cells were imaged using a 63×1.3 NA oil objective and processed with an Airyscan processing factor of 7.0. For analysis, Z-stacks were max-projected in FIJI. Images were background subtracted using a rolling ball correction. Hair cells with a FM 1–43 FX intensity more than twice the background intensity were scored as FM 1–43 FX positive.

## Zebrafish Ca$^{2+}$ imaging and behavior

For imaging of mechanosensitive Ca$^{2+}$ signals in neuromast hair bundles, *Tg(myo6b:memGCaMP6s)* animals were mounted on their side on Sylgard chamber in E3 embryo media containing 0.2% MESAB (*Lukasz and Kindt, 2018*). Larvae were then pinned and a solution containing α-bungarotoxin (125 µM, Tocris) was injected into the heart cavity for paralysis during imaging. After paralysis, larvae were immersed in extracellular imaging solution (in mM: 140 NaCl, 2 KCl, 2 CaCl$_2$, 1 MgCl$_2$, and 10 HEPES, pH 7.3, OSM 310+/-10). To image Ca$^{2+}$ signals in the hair bundles we used a Bruker Swept-Field

confocal system equipped with a Rolera EM-C2 CCD camera (QImaging, Surrey, Canada) and a Nikon CFI Fluor 60×1.0 NA water immersion objective as previously described (*Zhang et al., 2018a*). The system includes a band-pass 488/561 nm filter set (59904-ET, Chroma) and is controlled using Prairie View software (Bruker Corporation). A piezoelectric motor (PICMA P-882.11–888.11 series, PI Instruments) attached to the objective and used to acquire 5-plane Z-stacks every 0.5 μm using a 35 μm slit at a 50 Hz frame rate for a 10 Hz volume rate. A fluid-jet composed of a glass capillary attached to a pressure clamp system (HSPC-2-SB, ALA) was used to deliver a 500 ms anterior and posterior step stimulation to deflect hair bundles along their axis of sensitivity.

To quantify GCaMP6s fluorescent intensity changes during stimulation, we registered the Z-stacks and then average projected the Z-stacks into a single plane. We then loaded the projected images into FIJI and used the Time Series Analyzer V3 plugin to create circular ROIs with a ~1.7 μm diameter. ROIs were placed on the center of each individual bundle. We then measured and plotted change in the mean intensity ($\Delta F/F_0$) within the region during the recording period. The mean intensity within each ROI was computed for each hair bundle. A hair bundle with a signal magnitude (peak value of intensity change upon stimulation) above 10% $\Delta F/F_0$ was considered mechanosensitive. To create the AVG hair bundle GCaMP6s ($\Delta F/F_0$) per neuromast plot in *Figure 5—figure supplement 2C*, only traces from mechanosensitive cells were averaged.

A Zantiks MWP behavioral system was used to examine startle responses. The Zantiks system tracked and monitored behavioral responses via a built-in infrared camera at 30 frames per second. A 12-well plate was used to house larvae during behavioral analysis. Each well was filled with E3 and 1 larva. All fish were acclimated in the plate within the Zantiks chamber in dark for 15 min before each test. To induce startle, an integrated stepper motor was used to drive a vibration-induced startle response. A vibrational stimulus that triggered a maximal % of animals startling in controls without any tracking artifacts (due to the vibration) was used for our analyses. Each larva was presented with the vibrational stimulus 5 times with 100 s between trials. During the tracking and stimulation, a Cisco router connected to the Zantiks system was used to relay *x*, *y* coordinates of each larva every frame. To qualify as a startle response, a distance above four pixels or ~1.9 mm was required within two frames after stimulus onset. Animals were excluded from our analysis if no tracking data was recorded for the animal.

## Zebrafish statistics

All data shown are mean ± standard error of the mean (SEM). All experiments were performed on at least four animals and eight neuromasts and four cristae. Behavior analysis and Ca²⁺ imaging was repeated at least twice on two independent days. For in vivo work (calcium imaging and FM 1–43 labeling) fish were chosen at random and experimenter was blinded when possible. All replicates represented in graphs are biological and no samples were excluded. A power analysis was used to ensure that a sufficient sample size was used for our in vivo work. All statistical analysis was performed using GraphPad Prism 8.0. A D'Agostino and Pearson test was used to test data for normality. A one-way ANOVA with a Dunnett's correction for multiple comparisons was used to compare differences between siblings and controls regarding the % of mechanosensitive hair cells, number of hair cells, % of FM 1–43 positive hair cells, average FM 1–43 label. For proportion of animals startling and comparisons for FM 1–43 FX label, a Kruskal-Wallis test with a Dunnett's correction for multiple comparisons was used.

## Expression and purification of bacterially expressed *hs* CIB2, CIB3, TMC1-IL1, and TMC1-NT

DNA sequences encoding for *hs* CIB2, CIB3, TMC1-IL1, and TMC1-NT were subcloned into *NdeI* and *XhoI* sites of the pET21a vector. All DNA constructs were sequence verified. Protein fragments were expressed in *Escherichia coli* BL21 Rosetta (DE3) (Novagen) cells, which were cultured in LB or TB media, induced at $OD_{600}$~0.4–0.6 with 1 mM IPTG and grown at 20 °C or 30 °C for ~16 hr. Protein fragments for NMR were produced using the same cells cultured in M9 minimal media supplemented with MEM vitamin solution (Gibco) and 1 g/L ¹⁵NH₄Cl (Cambridge Isotopes) to produce uniformly labeled [U-¹⁵N] protein. Cells were lysed by sonication in denaturing buffer (20 mM Tris HCl, pH 7.5, 6 M guanidine hydrochloride, 10 mM CaCl₂, and 20 mM imidazole) or non-denaturing buffer (20 mM Tris HCl, pH 7.5, and 20 mM imidazole for *hs* TMC1-NT). The cleared lysates were loaded onto Ni-Sepharose (GE

Healthcare) and eluted with denaturing buffer or non-denaturing buffer supplemented with 500 mM imidazole. Denatured proteins were refolded using MWCO 2000 membranes (Spectra/Por 7) in dialysis buffer (400 mM Arg-HCl, 20 mM Tris-HCl, pH 7.5, 150 mM KCl, 50 mM NaCl, 2 mM $CaCl_2$, 2 mM DTT) for ~16 hr. Before starting refolding reactions (when applicable), elution solutions were diluted to ~0.5 mg/mL using the denaturing buffer, and then reduced by using 2 mM DTT in the diluted sample. Refolded and soluble proteins were concentrated using Vivaspin 20 or Amicon 15 centrifugal concentrators (10 kDa, 5 kDa, or 3 kDa molecular weight cutoff) and further purified on Superdex S75 columns (GE Healthcare) in SEC buffer (50 mM HEPES, pH 7.5, 126 mM KCl, 10 mM NaCl, 3 mM $CaCl_2$, 5 mM DTT). Proteins were concentrated to ~0.5 mL for NMR experiments.

## NMR

Purified NMR samples were at concentrations of ~1.4 mM, ~0.2 mM, ~0.8 mM, and ~0.2 mM for CIB2, CIB3, CIB2 + TMC1-IL1, and CIB3 + TMC1-IL1, respectively, in SEC buffer (50 mM HEPES, pH 7.5, 126 mM KCl, 10 mM NaCl, 3 mM $CaCl_2$, 5 mM DTT), supplemented with 10% (v/v) $D_2O$ and 0.2 mM 2,2-dimethyl-2-silapentane-5-sulphonate (DSS) for field-frequency locking and chemical shift referencing. Two-dimensional $^1H$-$^{15}N$ TROSY-HSQC spectra (*Nietlispach, 2005*) were recorded for each sample at 25 °C on a Bruker Avance III HD 800 MHz spectrometer equipped with a 5 mm TXI cryoprobe with z-gradients. Data were collected using 20–24 scans, 2048×280 datapoints, and spectral widths of 17482.518×4054.298 Hz in the $^1H$ and $^{15}N$ dimensions, respectively. Data were processed and visualized using NMRFx (*Norris et al., 2016*).

## Simulated systems

Models for monomeric *hs* TMC1 (residues 84–760; NCBI ID: NP_619636.2) and *hs* TMC2 (148–863; NCBI ID: NP_542789.2) in complex with full-length *hs* CIB2 (NCBI ID: NP_006374.1) and *hs* CIB3 (NCBI ID: NP_473454.1), as well as truncated dimeric models of *hs* TMC1 (residues 260–760; NCBI ID: NP_619636.2) and *hs* TMC2 (residues 324–863; NCBI ID: NP_542789.2) were generated using AF2 (*Evans et al., 2021*; *Jumper et al., 2021*). The TMC2 C-terminal end (residues 818–863) was truncated to match the C-terminal end of TMC1. Using VMD (*Humphrey et al., 1996*), a model of dimeric *hs* TMC1 + *hs* CIB3-$Ca^{2+}$ was constructed by aligning the monomeric *hs* TMC1 + *hs* CIB3 complex to the transmembrane region of the truncated dimeric *hs* TMC1 model. Coordinates for $Ca^{2+}$ ions were obtained by aligning the crystal structure of $Mg^{2+}$-bound *hs* CIB3 in complex with a *hs* TMC1-IL1 peptide (PDB code: 6WUD) (*Liang et al., 2021*) to the AF2 model of monomeric *hs* TMC1 + *hs* CIB3, then manually changing $Mg^{2+}$ for $Ca^{2+}$. This model was minimized for 1000 steps in vacuum using NAMD (*Phillips et al., 2005*) and the CHARMM36 force field (*Best et al., 2012*; *Huang and MacKerell, 2013*; *Vanommeslaeghe and MacKerell, 2015*) and served as a reference model for aligning other combinations of *hs* TMC1 or *hs* TMC2 with *hs* CIB2 or *hs* CIB3. All models were either embedded into pure POPC or mixed membrane lipid bilayers using CHARMM-GUI (*Jo et al., 2009*; *Jo et al., 2008*; *Jo et al., 2007*; *Lee et al., 2019*) as summarized in *Figure 8—source data 1*. The mixed membrane mimicked stereocilia membrane composition (*Zhao et al., 2012*). The protein-lipid system was then solvated using TIP3P explicit water and neutralized with 0.150 M KCl, representative of endolymph ionic concentration (*Bosher and Warren, 1978*).

## Simulations using NAMD

All-atom MD simulations were performed and analyzed using NAMD 2.13 (*Phillips et al., 2005*) and the CHARMM36 force field with CMAP corrections (*Best et al., 2012*; *Huang and MacKerell, 2013*; *Vanommeslaeghe and MacKerell, 2015*). For all systems, equilibrium simulations consisted of 2000 steps of minimization, 0.5 ns of dynamics with everything excluding lipid tails fixed/constrained, 0.5 ns of dynamics with harmonic constraints applied to the protein (1 kcal mol$^{-1}$ Å$^{-2}$), 1 ns of free dynamics in the *NpT* ensemble ($\gamma$ = 1 ps$^{-1}$), and up to 100 ns of free dynamics in the *NpT* ensemble ($\gamma$ = 0.1 ps$^{-1}$) prior to non-equilibrium production runs. A piston period of 200 fs and a damping timescale of 50 fs was used for pressure control and atomic coordinates were saved every 5 ps. All simulations were performed at 310 K and 1 atmosphere of pressure by using the Langevin thermostat and Nosé-Hoover Langevin pressure control. A 12 Å cutoff distance was employed with a force-based switching function starting at 10 Å. Periodic boundary conditions and the PME method were used to calculate long-range electrostatic interactions with a grid density greater than 1 Å$^{-3}$. An integration timestep of

2 fs with electrostatic interactions computed every other timestep was utilized for every simulation in tandem with the SHAKE algorithm for the constraint of hydrogen atoms. A constant electric field was applied to all atoms in non-equilibrium simulations with transmembrane potential (*Gumbart et al., 2012*).

## Simulations using Anton 2

We used Anton 2, a massively parallel special purpose machine for MD simulations (*Shaw et al., 2014*), to run simulations of *hs* TMC1 + *hs* CIB2-Ca$^{2+}$ in POPC (simulations S1d-g; *Figure 8—source data 1*), *hs* TMC1 + *hs* CIB2 in POPC (simulations S2d-2g; *Figure 8—source data 1*), *hs* TMC1 in POPC (simulations S3d-g; *Figure 8—source data 1*), and *hs* TMC1 + *hs* CIB2-Ca$^{2+}$ in a mixed membrane (simulations S4c-d; *Figure 8—source data 1*). Coordinates of the pre-equilibrated systems were converted to an Anton 2 compatible dms file format using the NAMDtoDMS.py script provided by the Pittsburgh Supercomputing Center. Simulations used CHARMM36 forcefields and were performed in the *NpT* ensemble (310 K, 1 atm) using the multigrator integration framework (*Lippert et al., 2013*) with a 2.5 fs integration timestep and saved every 120 ps. The MTK barostat and Nosé-Hoover thermostat was updated every 480 and 24 steps, respectively. Semi-isotropic pressure coupling was used. Long range electrostatic interactions were calculated using u-series electrostatics with a 64×64×64 grid. Van der Waals interactions were calculated with a cutoff of 12 Å. Transmembrane voltages were generated by applying a constant electric field to all atoms (*Gumbart et al., 2012*).

## Simulation analysis procedures and tools

BSA was computed in VMD by measuring SASA for both TMC and CIB monomers, then subtracting SASA for the complex. BSA for interacting molecules A and B were computed as $BSA_{AB} = 1/2 \left( SASA_A + SASA_B - SASA_{AB} \right)$. RMSD values were computed in VMD using Cα atoms and initial models as references, with alignments carried out using the corresponding subdomains when indicated. VMD was used to analyze trajectories, render molecular images, and create videos. Xmgrace and Gnuplot were used to generate plots. Sequences for CIB and TMC isoforms from various species were obtained from the NCBI protein database. Sequences were aligned utilizing the MUSCLE algorithm within Geneious (*Kearse et al., 2012*). Aligned sequences were loaded into JalView (*Waterhouse et al., 2009*) and colored based on a sequence identity threshold of 45%. Ionic currents were computed by counting the number of crossing, with conductance values estimated as in *Walujkar et al., 2021*.

## Resource availability

### Lead contact

Further information and requests for resources and reagents should be directed to the Lead Contacts, Drs. Marcos Sotomayor (sotomayor@uchicago.edu) and Zubair M. Ahmed (zmahmed@som.umaryland.edu).

### Materials availability

Materials generated in this study, including strains, plasmids, and clones, are freely available from the Lead Contact upon request.

## Acknowledgements

We acknowledge Sakina Rehman for assistance with animal work, Neal Taliwal for assistance with molecular cloning, Chunhua Yuan for assistance with initial nuclear magnetic resonance experiments, and Collin R Nisler and Harsha Mandayam Bharathi for assistance with simulations analyses. Simulations were carried out at the Ohio Supercomputer Center (OSC grants PAS1037 and PAA0217) and the Pittsburgh Supercomputer Center (PSC Anton 2 grant MCB150024P). Nuclear magnetic resonance data were acquired at the Ohio State University Campus Chemical Instrument Center. JSM was supported by an OSU/NIH training grant (T32 GM144293). This study was supported by NIH/NIDCD grants R01DC015271 to MS and R01DC012564 to ZMA, an Intramural Research Program Grant 1ZIADC000085-01 to KSK, and NIH grants UF1NS111695, R01DC018304, R01DC002390, R01DC018061 to KEC.

## Additional information

### Funding

| Funder | Grant reference number | Author |
|---|---|---|
| National Institute on Deafness and Other Communication Disorders | R01DC015271 | Marcos Sotomayor |
| National Institute on Deafness and Other Communication Disorders | R01DC012564 | Zubair M Ahmed |
| National Institute on Deafness and Other Communication Disorders | 1ZIADC000085-01 | Katie S Kindt |
| National Institutes of Health | UF1NS111695 | Kathleen E Cullen |
| National Institute on Deafness and Other Communication Disorders | R01DC018304 | Kathleen E Cullen |
| National Institute on Deafness and Other Communication Disorders | R01DC002390 | Kathleen E Cullen |
| National Institute on Deafness and Other Communication Disorders | R01DC018061 | Kathleen E Cullen |
| Ohio Supercomputer Center | PAS1037 | Marcos Sotomayor |
| Ohio Supercomputer Center | PAA0217 | Marcos Sotomayor |
| Pittsburgh Supercomputer Center | MCB150024P | Marcos Sotomayor |

The funders had no role in study design, data collection and interpretation, or the decision to submit the work for publication.

### Author contributions

Arnaud PJ Giese, Carried out all audiometric recordings in mice, immunolocalization studies, cochlear and vestibular explant cultures, FM 1-43 uptake experiment on mouse tissues, RT-qPCRs, exploratory testing, Cib2 and Cib3 mouse phenotyping and genotyping, cloning of CIB2 and CIB3 in pET21a vectors. Contributed to data analysis. Wrote the manuscript; Wei-Hsiang Weng, Did the cloning and mutagenesis of CIB and TMC constructs, protein expression and purification for NMR experiments, as well as model building and molecular dynamics simulations. Contributed to data analysis. Wrote the manuscript; Katie S Kindt, Conceived the studies. Conducted zebrafish FM 1-43, FM 4-64, FM 1-43FX labeling, Ca2+ imaging and immunostaining. Contributed to data analysis. Wrote the manuscript; Hui Ho Vanessa Chang, Performed VOR, OKN measurements and vestibular function evaluation of mice. Contributed to data analysis; Jonathan S Montgomery, Did NMR experiments and NMR data processing. Contributed to data analysis; Evan M Ratzan, Performed FM 1-43 uptake experiments in mice. Contributed to data analysis; Alisha J Beirl, Created the *cib2* and *cib3* zebrafish mutants. Contributed to data analysis; Roberto Aponte Rivera, Did the zebrafish behavioral studies. Contributed to data analysis; Jeffrey M Lotthammer, Did model building. Contributed to data analysis; Sanket Walujkar, Did model building. Contributed to data analysis; Mark P Foster, Supervised NMR experiments and project. Contributed to data analysis; Omid A Zobeiri, Performed VOR, OKN measurements and vestibular function evaluation of mice. Contributed to data analysis; Jeffrey R Holt, Analyzed the mouse FM 1-43 experiments data, provided supervision and resources. Contributed to data analysis; Saima Riazuddin, Supervised the project, reviewed and interpreted the results, provided resources. Contributed to data analysis; Kathleen E Cullen, Contributed to data analysis. Wrote the manuscript; Marcos Sotomayor, Conceived the studies. Supervised project and assisted with cloning,

strategies for protein refolding, model building, and simulations. Contributed to data analyses. Wrote the manuscript; Zubair M Ahmed, Conceived the studies. Supervised the project, reviewed and interpreted the results, provided resources. Contributed to data analysis. Wrote the manuscript

### Author ORCIDs
Arnaud PJ Giese ⓘ https://orcid.org/0000-0001-7228-9542
Wei-Hsiang Weng ⓘ https://orcid.org/0000-0001-8146-930X
Katie S Kindt ⓘ https://orcid.org/0000-0002-1065-8215
Jonathan S Montgomery ⓘ https://orcid.org/0000-0003-2463-6671
Roberto Aponte Rivera ⓘ https://orcid.org/0000-0002-1767-527X
Sanket Walujkar ⓘ https://orcid.org/0000-0002-5892-4578
Mark P Foster ⓘ https://orcid.org/0000-0001-9645-7491
Jeffrey R Holt ⓘ https://orcid.org/0000-0002-7182-8011
Kathleen E Cullen ⓘ https://orcid.org/0000-0002-9348-0933
Marcos Sotomayor ⓘ http://orcid.org/0000-0002-3333-1805
Zubair M Ahmed ⓘ https://orcid.org/0000-0003-2914-4502

### Ethics
All animal procedures were approved by the Institutional Animal Care and Use Committees (IACUCs) at University of Maryland (protocol #0420002) and at Harvard Medical School (protocol #00001240). Zebrafish work at the National Institute of Health (NIH) was approved by the Animal Use Committee at the NIH under animal study protocol #1362-13.

Reviewer #1 (Public Review): https://doi.org/10.7554/eLife.89719.3.sa1
Reviewer #2 (Public Review): https://doi.org/10.7554/eLife.89719.3.sa2
Author response https://doi.org/10.7554/eLife.89719.3.sa3

## Additional files

### Supplementary files
MDAR checklist

### Data availability
Data generated or analyzed during this study are included in the manuscript, supporting files and in Dryad and Zenodo.

The following datasets were generated:

| Author(s) | Year | Dataset title | Dataset URL | Database and Identifier |
|---|---|---|---|---|
| Kindt K | 2024 | Data from: Complexes of vertebrate TMC1/2 and CIB2/3 proteins form hair-cell mechanotransduction cation channels | https://doi.org/10.5061/dryad.4tmpg4fkw | Dryad Digital Repository, 10.5061/dryad.4tmpg4fkw |
| Kindt K | 2024 | Data from: Complexes of vertebrate TMC1/2 and CIB2/3 proteins form hair-cell mechanotransduction cation channels | https://doi.org/10.5281/zenodo.13850420 | Zenodo, 10.5281/zenodo.13850420 |

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
