## [Editor Report · eLife Assessment]

This paper, on the role of calcium and integrin-binding protein 2 and 3 in the hair-cell in the mechano-electrical transduction (MET) apparatus, is a mix of confirmatory studies with new and potentially **important** data. Some parts, such as zebrafish studies, the modelling and simulations, are regarded as necessary and **convincing**. Other parts of the paper do not have the same novelty. Both Liang et al. (2021) and Wang et al. (2023) had previously demonstrated a role for CIB2/CIB3 in auditory and vestibular cells in mice. Moreover, there are also data in Riazuddin et al. (2012) paper that demonstrates the importance of CIB2 in zebrafish and *Drosophila*. Breaking the manuscript up to focus on specific aspects of the problem might alleviate the limitations of this multi-faceted study.

---

## [Referee Report · Reviewer #1 (Public Review)]

Revised Public Review

This reviewed preprint is essentially three papers combined into one-one paper focused on the role of CIB2/CIB3 in vestibular hair cells, one on the role of CIB2/CIB3 in zebrafish, and one on structural modeling of a CIB2/3 and TMC1/2 complex. The authors try to combine the three parts with the overarching theme of demonstrating that CIB2/3 play a functionally conserved role across species and hair cell types. It is important to note that many of the basic results from the mouse have already been reported by other groups in Liang et al. (2021) and Wang et al. (2023).

That said, their demonstration of the importance of CIB2 and CIB3 in zebrafish hair cell function is novel. The results largely coincide with what is seen in the mouse-they are both important, with stimulus-dependent Ca2+ entry reduced more in cib2 KOs than in cib3 KOs, and the cib2;cib3 showing the greatest impact. Interestingly, cib2 is uniquely localized in and important for specific hair cell types in the neuromast and crista.

The last part of the manuscript also offers significant new findings. Here structural studies (AlphaFold 2 modeling, NMR structure determination, and molecular dynamics simulations) brings us closer to the structure of the mammalian TMCs, alone and in complex with the CIB proteins. Moreover, the structural work supports the assignment of the TMC pore to alpha helices 4-7.

In summary, while this reviewed preprint has some data that replicate data from publications from other labs, it provides a comprehensive look at the CIB family in hair cells, especially in vestibular hair cells.

---

## [Referee Report · Reviewer #2 (Public Review)]

The paper by Giese and coworkers is quite an intense reading. The manuscript is packed with data pertaining to very different aspects of MET apparatus function, scales, and events. I have to praise the team that combined molecular genetics, biochemistry, NMR, microscopy, functional physiology, in-vivo tests for vestibulo-ocular reflexes, and other tests for vestibular dysfunction with molecular modeling and simulations. The authors nicely show the way CIBs are associated with TMCs to form functional MET channels. The authors clarify the specificity of associations and elucidate the functional effects of the absence of specific CIBs and their partial redundancy.

Comments on revised version:

I appreciate the author's effort to address my comments. The revised paper 'Complexes of vertebrate TMC1/2 and CIB2/3 proteins 1 form hair-cell mechanotransduction cation channels' by Giese and coworkers is definitely cleaner but remains a compendium of related but very uneven parts. By saying 'uneven,' I mean that the grounding of the experimental and computational parts is different, and the firmness of conclusions, respectively, is not matched.

My conclusion is that this is a great collaborative project. However, in its present form, different components pull the emphasis in several directions with little cross-talk. It is worth splitting into two papers.

---

## [Author Response]

The following is the authors’ response to the original reviews.

**Public Reviews:**

**Reviewer #1 (Public Review):**
This reviewed preprint is a bit of Frankenstein monster, as it crams together three quite different sets of data. It is essentially three papers combined into one-one paper focused on the role of CIB2/CIB3 in VHCs, one on the role of CIB2/CIB3 in zebrafish, and one on structural modeling of a CIB2/3 and TMC1/2 complex. The authors try to combine the three parts with the overarching theme of demonstrating that CIB2/3 play a functionally conserved role across species and hair cell types, but given the previous work on these proteins, especially Liang et al. (2021) and Wang et al. (2023), this argument doesn't work very well. My sense is that the way the manuscript is written now, the sum is less than the individual parts, and the authors should consider whether the work is better split into three separate papers.

We appreciate the frank evaluation of our work and point out that combining structural with functional data from mouse and zebrafish offers a comprehensive view of the role played by TMC1/TMC2 and CIB2/3 complexes in hair-cell mechanotransduction. We believe that readers will benefit from this comprehensive analyses.

The most important shortcoming is the novelty of the work presented here. In line 89 of the introduction the authors state "However, whether CIB2/3 can function and interact with TMC1/2 proteins across sensory organs, hair-cell types, and species is still unclear." They make a similar statement in the first sentence of the discussion and generally use this claim throughout the paper as motivation for why they performed the experiments. Given the data presented in the Liang et al. (2021) and Wang et al. (2023 papers), however, this statement is not well supported. Those papers clearly demonstrate a role for CIB2/CIB3 in auditory and vestibular cells in mice. Moreover, there is also data in Riazuddin et al. (2012) paper that demonstrates the importance of CIB2 in zebrafish and *Drosophila*. I think the authors are really stretching to describe the data in the manuscript as novel. Conceptually, it reads more as solidifying knowledge that was already sketched out in the field in past studies.

We note that work on mouse and fish CIB knockouts in our laboratories started over a decade ago and that our discoveries are contemporary to those recently presented by Liang et al., 2021 and Wang et al., 2023, which we acknowledge, cite, and give credit as appropriate. We also note that work on fish knockouts and on fish *Cib3* is completely novel. Nevertheless, the abstract text “Whether these interactions are functionally relevant across mechanosensory organs and vertebrate species is unclear” has been replaced by “These interactions have been proposed to be functionally relevant across mechanosensory organs and vertebrate species.”; and the introduction text “However, whether CIB2/3 can function and interact with TMC1/2 proteins across sensory organs, hair-cell types, and species is still unclear” has been replaced by “However, additional evidence showing that CIB2/3 can function and interact with TMC1/2 proteins across sensory organs, hair-cell types, and species is still needed.”. The work by Wang et al., 2023 is immediately discussed after the first sentence in the discussion section and the work by Liang et al., 2021 is also cited in the same paragraph. We believe that changes in abstract and introduction along with other changes outlined below put our work in proper context.

There is one exception, however, and that is the last part of the manuscript. Here structural studies (AlphaFold 2 modeling, NMR structure determination, and molecular dynamics simulations) bring us closer to the structure of the mammalian TMCs, alone and in complex with the CIB proteins. Moreover, the structural work supports the assignment of the TMC pore to alpha helices 4-7.

Thanks for the positive evaluation of this work.

**Reviewer #2 (Public Review):**
The paper 'Complexes of vertebrate TMC1/2 and CIB2/3 proteins 1 form hair-cell mechanotransduction cation channels' by Giese and coworkers is quite an intense reading. The manuscript is packed with data pertaining to very different aspects of MET apparatus function, scales, and events. I have to praise the team that combined molecular genetics, biochemistry, NMR, microscopy, functional physiology, in-vivo tests for vestibulo-ocular reflexes, and other tests for vestibular dysfunction with molecular modeling and simulations. The authors nicely show the way CIBs are associated with TMCs to form functional MET channels. The authors clarify the specificity of associations and elucidate the functional effects of the absence of specific CIBs and their partial redundancy.

We appreciate the positive evaluation of our work and agree with the reviewer in that the combination of data obtained using various techniques in vivo and in silico provide a unique view on the role played by CIB2 and CIB3 in hair-cell mechanotransduction.

**Reviewer #3 (Public Review):**
This study demonstrates that from fish to mammals CIB2/3 is required for hearing, revealing the high degree of conservation of CIB2/3 function in vertebrate sensory hair cells. The modeling data reveal how CIB2/3 may affect the conductance of the TMC1/2 channels that mediate mechanotransduction, which is the process of converting mechanical energy into an electrical signal in sensory receptors. This work will likely impact future studies of how mechanotransduction varies in different hair cell types.One caveat is that the experiments with the mouse mutants are confirmatory in nature with regard to a previous study by Wang et al., and the authors use lower resolution tools in terms of function and morphological changes. Another is that the modeling data is not supported by electrophysiological experiments, however, as mentioned above, future experiments may address this weakness.

We thank the reviewer for providing positive feedback and for highlighting caveats that can and will be addressed by future experiments.

**Reviewer #1 (Recommendations For The Authors):**
Lines 100-101. Please temper this statement, as FM1-43 is only a partial proxy for MET.

The original text has been modified to: “In contrast to auditory hair cells, we found that the vestibular hair cells in *Cib2KO/KO* mice apparently have MET. We assessed MET via uptake of FM 1-43 (Figure 1A), a styryl dye that mostly permeates into hair cells through functional MET channels (Meyers et al., 2003), indicating that there may be another CIB protein playing a functionally redundant role.”

Lines 111-113. These data do not fully match up with the Kawashima et al. (2011) data. Please discuss.

We have modified the text to better report the data: “*Tmc2* expression increases during development but remains below *Tmc1* levels in both type 1 and type 2 hair cells upon maturation (Figure 1C).”

Lines 125-126. The comparison in 2A-B is not described correctly for the control. The strain displayed is Cib2^+/+;Cib3^KO/KO (not wild-type). Show the Cib2^+/+;Cib3^+/+ if you are going to refer to it (and is this truly Cib2^+/+;Cib3^+/+ from a cross or just the background strain?).

Thanks for pointing this out. To avoid confusion, we have revised the sentence as follow: “We first characterized hearing function in *Cib3KO/KO* and control littermate mice at P16 by measuring auditory-evoked brainstem responses (ABRs). Normal ABR waveforms and thresholds were observed in *Cib3KO/KO* indicating normal hearing.”

Lines 137-140. Did you expect anything different? This is a trivial result, given the profound loss of hearing in the Cib2^KO/KO mice.

We did not expect anything different and have deleted the sentence: “Furthermore, endogenous CIB3 is unable to compensate for CIB2 loss in the auditory hair cells, perhaps due to extremely low expression level of CIB3 in these cells and the lack of compensatory overexpression of CIB3 in the cochlea of *Cib2KO/KO* mice (Giese et al., 2017).”

Lines 194-196. But what about Cib2^KO/KO; Isn't the conclusion that the vestibular system needs either CIB2 or CIB3?

Yes, either CIB2 or CIB3 can maintain normal vestibular function. A prior study by Michel et al., 2017, has evaluated and reported intact vestibular function in *Cib2KO/KO* mice.

Lines 212-214. Yes. This is a stronger conclusion than the one earlier.

We have revised the sentence as follow: “Taken together, these results support compulsory but functionally redundant roles for CIB2 and CIB3 in the vestibular hair cell MET complex.”

Lines 265-267. I'm not sure that I would state this conclusion here given that you then argue against it in the next paragraph.

We have modified this statement to make the conclusions clearer and more consistent between the two paragraphs. The modified text reads: “Thus, taken together the results of our FM 1-43 labeling analysis are consistent with a requirement for both Cib2 and Cib3 to ensure normal MET in all lateral-line hair cells.”

Line 277. I would be more precise and say something like "and sufficiently fewer hair cells responded to mechanical stimuli and admitted Ca2+..."

We have modified the text as requested: “We quantified the number of hair bundles per neuromast with mechanosensitive Ca2+ responses, and found that compared to controls, significantly fewer cells were mechanosensitive in *cib2* and *cib2*;*cib3* mutants (Figure 5-figure supplement 2A, control: 92.2 ± 2.5; *cib2*: 49.9 ± 5.8, *cib2;cib3:* 19.0 ± 6.6, p > 0.0001).”

Line 278 and elsewhere. It doesn't make sense to have three significant digits in the error. I would say either "92.2 {plus minus} 2.5" or "92 {plus minus} 2."

Edited as requested.

Lines 357-358. Move the reference to the figure to the previous sentence, leaving the "(Liang et al., 2021) juxtaposed to its reference (crystal structure). Otherwise, the reader will look for crystal structures in Figure 7-figure supplements 1-5.

Text has been edited as requested: “The intracellular domain linking helices a2 and a3, denoted here as IL1, adopts a helix-loop-helix with the two helices running parallel to each other and differing in length (Figure 7-figure supplements 1-5). This is the same fold observed in its crystal structure in complex with CIB3 (Liang et al., 2021), which validated the modeling approach.”

Line 450. What other ions were present besides K+? I assume Cl- or some other anion.What about Na+ or Ca+? It's hard to evaluate this sentence without that information.

Systems have 150 mM KCl and CIB-bound Ca2+ when indicated (no Na+ or free Ca2+). This is now pointed out when the models are described first: “These models were embedded in either pure POPC or stereocilia-like mixed composition bilayers and solvated (150 mM KCl) to …”. The sentence mentioned by the reviewer has also been modified: “In systems with pure POPC bilayers we observed permeation of K+ in either one or both pores of the TMC1 dimer, with or without CIB2 or CIB3 and with or without bound Ca2+, despite the presence of Cl- (150 mM KCl).”

Lines 470-472. These results suggest that the maximum conductance of TMC1 > TMC2. How do these results compare with the Holt and Fettiplace data?

Thanks for pointing this out. A comparison would be appropriate and has been added: “We also speculate that this is due to TMC2 having an intrinsic lower singlechannel conductance than TMC1, as has been suggested by some experiments (Kim et al., 2013), but not others (Pan et al., 2013). It is also possible that our TMC2 model is not in a fully open conformation, which can only be reached upon mechanical stimulation.”

Line 563. Yes, the simulations only allow you to say that the interaction is stable for at least microseconds. However, the gel filtration experiments suggest that the interaction is stable for much longer. Please comment.

Thank you for pointing this out. We agree with this statement and modified the text accordingly: “Simulations of these models indicate that there is some potential preferential binding of TMC1 and TMC2 to CIB3 over CIB2 (predicted from BSA) and that TMC + CIB interactions are stable and last for microseconds, with biochemical and NMR experiments showing that these interactions are stable at even longer timescales.”

Figure 3. Please use consistent (and sufficiently large to be readable) font size.

Figure has been updated.

Figure 4. Magnification is too low to say much about bundle structure.

The reviewer is right – we cannot evaluate bundle structure with the images shown in Figure 4. Our goal was to determine if the vestibular hair cells had been degenerated in the absence of CIB2/3 and Figure 4 panel A data reveals intact hair cells. We changed the text “High-resolution confocal imaging did not reveal any obvious vestibular hair cell loss and hair bundles looked indistinguishable from control in *Cib2KO/KO;Cib3KO/KO* mice (Figure 4A).” to “High-resolution confocal imaging did not reveal any obvious vestibular hair cell loss in *Cib2KO/KO;Cib3KO/KO* mice (Figure 4A).” to avoid any confusions.

**Reviewer #2 (Recommendations For The Authors):**
Some datasets presented here can be published separately. Although I understand that the field is developing fast and there is no time to sort and fit the data by category or scale, everything needs to be published together and quickly.I have no real questions about the data on the functional association of CIB2 and 3 with TMC 1 and 2 in mouse hair cells as well as association preferences between their homologs in zebrafish. The authors have shown a clear differentiation of association preferences for CIB2 and CIB3 and the ability to substitute for each other in cochlear and vestibular hair cells. The importance of CIB2 for hearing and CIB3 for vestibular function is well documented. The absence of the startle response in cib2/3 negative zebrafish is a slight variation from what was observed in mice where CIB2 is sufficient for hearing. The data look very solid and show an overall structural and functional conservation of these complexes throughout vertebrates. The presented models look plausible, but of course, there is a chance that they will be corrected/improved in the future.

Thanks for appreciating the significance of our study.

Regarding NMR, there is indeed a large number of TROSY peaks of uniformly labeled CIB2 undergoing shifts with sequential additions of the loop and the N-terminal TMC peptides. Something is going on. The authors may consider a special publication on this topic when at least partial peak assignments are established.

We are continuing our NMR studies of CIB and TMC interactions and plan to have follow up studies.

After reading the manuscript, I may suggest four topics for additional discussion.(1) Maybe it is obvious for people working in the field, but for the general reader, the simulations performed with and without Ca2+ come out of the blue, with no explanation. The authors did not mention clearly that CIB proteins have at least two functional EF-hand (EF-hand-like) motifs that likely bind Ca2+ and thereby modulate the MET channel.

This is a good point. We have modified the introductory text to include: “CIB2 belongs to a family of four closely related proteins (CIB1-4) that have partial functional redundancy and similar structural domains, with at least two Ca2+/Mg2+-binding EF-hand motifs that are highly conserved for CIB2/3 (Huang et al., 2012).”

If the data on affinities for Ca2+, as well as Ca2+-dependent propensity for dimerization and association with TMC exist, they should be mentioned for CIB2 and CIB3 and discussed.

To address this, we have added the following text to the discussion: “How TMC + CIB interactions depend on Ca2+ concentration may have important functional implications for adaptation and hair cell mechanotransduction. Structures of CIB3 and worm CALM-1, a CIB2 homologue, both bind divalent ions via EF-hand motifs proximal to their C-termini (Jeong et al., 2022; Liang et al., 2021). Reports on CIB2 affinities for Ca2+ are inconsistent, with _K_D values that range from 14 µM to 0.5 mM (Blazejczyk et al., 2009; Vallone et al., 2018). Although qualitative pull-down assays done in the presence or the absence of 5 mM CaCl2 suggest that the TMC1 and CIB2 interactions are Ca2+independent (Liang et al., 2021), strength and details of the CIB-TMC-IL1 and CIB-TMCNT contacts might be Ca2+-dependent, especially considering that Ca2+ induces changes that lead to exposure of hydrophobic residues involved in binding (Blazejczyk et al., 2009).”

Also, it is not clearly mentioned in the figure legends whether the size-exclusion experiments or TROSY NMR were performed in the presence of (saturating) Ca2+ or not. If the presence of Ca2+ is not important, it must be explained.

Size exclusion chromatography and NMR experiments were performed in the presence of 3 mM CaCl2. We have indicated this in appropriate figure captions as requested, and also mentioned it in the discussion text: “Interestingly, the behavior of CIB2 and CIB3 in solution (SEC experiments using 3 mM CaCl2) is different in the absence of TMC1-IL1.” and “Moreover, our NMR data (obtained using 3 mM CaCl2) indicates that TMC1-IL1 + CIB2 is unlikely to directly interact with CIB3.”

(2) Speaking about the conservation of TMC-CIB structure and function, it would be important to compare it to the *C. elegans* TMC-CALM-1 structures. Is CALM-1, which binds Ca2+ near its C-terminus, homologous or similar to CIBs?

This is an important point. To address it, we have added the following text in the discussion: “Remarkably, the AF2 models are also consistent with the architecture of the nematode TMC-1 and CALM-1 complex (Jeong et al., 2022), despite low sequence identity (36% between human TMC1 and worm TMC-1 and 51% between human CIB2 and worm CALM-1). This suggests that the TMC + CIB functional relationship may extend beyond vertebrates.” We also added: “How TMC + CIB interactions depend on Ca2+ concentration may have important functional implications for adaptation and hair cell mechanotransduction. Structures of CIB3 and worm CALM-1, a CIB2 homologue, both bind divalent ions via EF-hand motifs proximal to their C-termini (Jeong et al., 2022; Liang et al., 2021).”

Additionally, superposition of CALM-1 (in blue) from the TMC-1 complex structure (PDB code: 7usx; Jeong et al., 2022) with one and our initial human CIB2 AF2 models (in red) show similar folds, notably in the EF-hand motifs of CALM-1 and CIB2 (Author response image 1).

**Author response image 1. sa3fig1:** Superposition of CALM-1 structure blue; Jeong et al., 2022) and AlphaFold 2 model of CIB2 (red). Calcium ions are shown as green spheres.

(1) Based on simulations, CIBs stabilize the cytoplasmic surfaces of the dimerized TMCs.The double CIB2/3 knock-out, on the other hand, clearly destabilizes the morphology of stereocilia and leads to partial degeneration. One question is whether the tip link in the double null forms normally and whether there is a vestige of MET current in the beginning. The second question is whether the stabilization of the TMC's intracellular surface has a functional meaning. I understand that not complete knock-outs, but rather partial loss-of-function mutants may help answer this question. The reader would be impatient to learn what process most critically depends on the presence of CIBs: channel assembly, activation, conduction, or adaptation. Any thoughts about it?

These are all interesting questions, although further investigations would be needed to understand CIB’s role on channel assembly, activation, conduction, and adaption. We have added to the discussion text: “Further studies should help provide a comprehensive view into CIB function in channel assembly, activation, and potentially hair-cell adaption.”

(2) The authors rely on the permeation of FM dyes as a criterion for normal MET channel formation. What do they know about the permeation path a 600-800 Da hydrophobic dye may travel through? Is it the open (conductive) or non-conductive channel? Do ions and FM dyes permeate simultaneously or can this be a different mode of action for TMCs that relates them to TMEM lipid scramblases? Any insight from simulations?

We are working on follow-up papers focused on elucidating the permeation mechanisms of aminoglycosides and small molecules (such as FM dyes) through TMCs as well as its potential scramblase activity.

**Reviewer #3 (Recommendations For The Authors):**
Introduction:The rationale and context for determining whether Cib2 and Cib3 proteins are essential for mechanotransduction in zebrafish hair cells is completely lacking in the introduction. All background information about what is known about the MET complex in sensory hair cells focuses on work done with mouse cochlear hair cells without regard to other species. This is especially surprising as the third author uses zebrafish as an animal model and makes major contributions to this study, addressing the primary question posed in the introduction. Instead, the authors relegate this important information to the results section. Moreover, not mentioning the Jeong 2022 study when discussing the Liang 2021 findings is odd considering that the primary question is centered on CIB2 and TMC1/2 in other species.

Thank you for pointing this out. We now discuss and reference relevant background on the MET complex in zebrafish hair cells in the introduction. We added: “In zebrafish, Tmcs, Lhfpl5, Tmie, and Pcdh15 are also essential for sensory transduction, suggesting that these molecules form the core MET complex in all vertebrate hair cells (Chen et al., 2020; Erickson et al., 2019, 2017; Ernest et al., 2000; Gleason et al., 2009; Gopal et al., 2015; Maeda et al., 2017, 2014; Pacentine and Nicolson, 2019; Phillips et al., 2011; Seiler et al., 2004; Söllner et al., 2004).”. We also added: “In zebrafish, knockdown of *Cib2* diminishes both the acoustic startle response and mechanosensitive responses of lateral-line hair cells (Riazuddin et al., 2012).”

Discussion:The claim that mouse vestibular hair cells in the double KO are structurally normal is not well supported by the images in Fig. 4A and is at odds with the findings by Wang et al., 2023. More discussion about the discrepancy of these results (instead of glossing over it) is warranted. The zebrafish image of the hair bundles in the zebrafish cib2/3 double knockout also appear abnormal, i.e. somewhat thinner. These results are consistent with Wang et al., 2023. Is it the case that neither images (mouse and fish) are representative? Unfortunately, the neuromast hair bundles in the double mutant are not shown, so it is difficult to draw a conclusion.

The reviewer is right – we cannot evaluate mouse hair-cell bundle structure with the images shown in Figure 4. Our goal was to determine if the vestibular hair cells had been degenerated in the absence of CIB2/3 and Figure 4 panel A data reveals intact hair cells. We changed the text “High-resolution confocal imaging did not reveal any obvious vestibular hair cell loss and hair bundles looked indistinguishable from control in *Cib2KO/KO;Cib3KO/KO* mice (Figure 4A).” to “High-resolution confocal imaging did not reveal any obvious vestibular hair cell loss in *Cib2KO/KO;Cib3KO/KO* mice (Figure 4A).” to avoid any confusions. In addition, we have changed the discussion as follows: “We demonstrate that vestibular hair cells in mice and zebrafish lacking CIB2 and CIB3 are not degenerated but have no detectable MET, assessed via FM 1-43 dye uptake, at time points when MET function is well developed in wild-type hair cells.”

In the discussion, the authors mention that Shi et al showed differential expression with cib2/3 in tall versus short hair cells of zebrafish cristae. However, there is no in situ data in the Shi study for cib2 and cib3. Instead, Shi et al show in situs for zpld1a and cabp5b that mark these cell types in the lateral crista. The text is slightly misleading and should be changed to reflect that UMAP data support this conclusion.

We have removed reference to cib2/3 zebrafish differential expression from our discussion. It is true that this differential expression has only been inferred by UMAP and not in situ data.

It should be noted that the acoustic startle reflex is mediated by the saccule in zebrafish, which does not possess layers of short and tall hair cells, but rather only has one layer of hair cells. Whether saccular hair cells can be regarded as strictly 'short' hair cell types remains to be determined. In this paragraph of the discussion, the authors are confounding their interpretation by not being careful about which endorgan they are discussing (line 521). In fact, there is a general error in the manuscript in referring to vestibular organs without specifying what is shown. The cristae in zebrafish do not participate in behavioral reflexes until 25 dpf and they are not known to synapse onto the Mauthner cell, which mediates startle reflexes.

Thank you for pointing out these issues. We now state in the results that the startle reflex in zebrafish relies primarily on the saccule. In the discussion we now focus mainly on short and tall hair cells of the crista. We also outline again in the discussion that the saccule is required for acoustic startle and the crista are for angular acceleration.

Minor points:Lines 298-302: The Zhu reference is not correct (wrong Zhu author). The statement on the functional reliance on Tmc2a versus Tmc1/2b should be referenced with Smith et al., 2020 and the correct Zhu 2021 study from the McDermott lab. Otherwise, the basis for the roles of the Tmcs in the cartoon in panel 6E is not clear.

Thanks for pointing out this oversight. We have updated the reference.

Line 548 should use numbers to make the multiple points, otherwise, this sentence is long and awkward.

The sentence has been re-arranged to make it shorter and to address another point raised by referees: “Structural predictions using AF2 show conserved folds for human and zebrafish proteins, as well as conserved architecture for their protein complexes. Predictions are consistent with previous experimentally validated models for the TMC1 pore (Ballesteros et al., 2018; Pan et al., 2018), with the structure of human CIB3 coupled to mouse TMC1-IL1 (Liang et al., 2021), and with our NMR data validating the interaction between human TMC1 and CIB2/3 proteins. Remarkably, the AF2 models are also consistent with the architecture of the nematode TMC-1 and CALM-1 complex (Jeong et al., 2022), despite low sequence identity (36% between human TMC1 and worm TMC-1 and 51% between human CIB2 and worm CALM-1). This suggests that the TMC + CIB functional relationship may extend beyond vertebrates.”

Suggested improvements to the figures:In general, some of the panels are so close together that keys or text for one panel look like they might belong to another. Increasing the white space would improve this issue.

Figure 3 has been adjusted as requested, Figure 7 has been split into two (Figure 7 and Figure 8) to make them more readable and to move data from the supplement to the main text as requested below.

Fig1A. The control versus the KO images look so different that this figure fails to make the point that FM labeling is unaffected. The authors should consider substituting a better image for the control. It is not ideal to start off on a weak point in the first panel of the paper.

We agree and have updated Figure 1 accordingly.

Fig1C. It is critical to state the stage here. Also P12?

scRNA-seq data are extracted from Matthew Kelley’s work and are a combination of P1, P12 and P100 utricular hair cells as following: Utricular hair cells were isolated by flow cytometry from 12- and 100-day old mice. Gene expression was then measured with scRNA-seq using the 10x platform. The data were then combined with a previously published single cell data set (samples from GSE71982) containing utricular hair cells isolated at P1. This dataset shows gene expression in immature vs mature utricular hair cells. The immature hair cells consist of a mixture of type I and type II cells.

Fig1D. This schematic is confusing. The WT and KO labels are misplaced and the difference between gene and protein diagrams is not apparent. Maybe using a different bar diagram for the protein or at least adding 'aa' to the protein diagrams would be helpful.

Sorry for the confusion. We have revised panel 1D to address these concerns.

Fig1E. Would be good to add 'mRNA' below the graph.

Done. We have added “mRNA fold change on the Y-axis” label.

Fig2C and D. Why use such a late-stage P18 for the immunohistochemistry?

Data presented in panel 2C are from P5 explants kept 2 days in vitro. For panel 2D, P18 is relevant since ABR were performed at P16 and hair cell degeneration in CIB2 mutants as previously described occurs around P18-P21.

Fig3A. Why isn't the cib2-/- genotype shown?

Data on cib2-/- mutant mice have already been published and no vestibular deficits have been found. See Giese *et al*., 2017 and Michel *et al*., 2017

Fig3F. Does this pertain to the open field testing? It would make sense for this panel to be associated with those first panels.

Figure 3 has been updated as requested.

Fig4A. Which vestibular end organ? Are these ampullary cells? (Same question for 4B.) The statement in the text about 'indistinguishable' hair bundles is not supported by these panels. There appears to be an obvious difference here--the hair bundles look splayed in the double KO. Either the magnification of the images is not the same or the base of the bundles is wider in the double KO as well. This morphology appears to be at odds with results reported by Wang et al., 2023.

The vestibular end organs shown in Figure 4A are ampullae. Magnifications are consistent across all the panels. While reviewer might be right regarding the hair bundle morphology, SEM data would be the best approach to address this point. Unfortunately, we currently do not have such data and we believe that only vestibular hair loss can be addressed using IF images. Thus, we are only commenting on the absence of obvious vestibular haircell loss in the double KO mutants.

Fig4C. To support the claim that extrastriolar hair cells in the Cib3-/- mice are less labeled with FM dye it would be necessary to at least indicate the two zones but also to quantify the fluorescence. One can imagine that labeling is quite variable due to differences in IP injection.

The two zones have been outlined in Figure 4C as requested.

Fig5. Strangely the authors dedicate a third of Figure 1 to describing the mouse KO of Cib3, yet no information is given about the zebrafish CRISPR alleles generated for this study. There is nothing in the results text or in this figure. At least one schematic could be added to introduce the fish alleles and another panel of gEAR information about cib2 and cib3 expression to help explain the neuromast data as was done in Fig1C.

We have added a supplemental figure (Figure 5-figure Supplement 1) that outlines where the zebrafish *cib2* and *cib3* mutations are located. We also state in the results additional information regarding these lesions. In addition, we provide context for examining cib2/3 in zebrafish hair cells by referencing published data from inner ear and lateral line scRNAseq data in the results section.

Absolutely nitpicky here, but the arrow in 5H may be confused for a mechanical stimulus.

The arrow in 5H has been changed to a dashed line.

Why not include the data from the supplemental figure at the end of this figure?

The calcium imaging data in the supplement could be included in the main figure but it would make for a massive figure. In eLife supplements can be viewed quite easily online, next to the main figures.

Fig6. The ampullary hair bundles look thinner in 6I. Is this also the case for double KO neuromast bundles? Such data support the findings of Wang et al., 2023.

We did not quantify the width of the hair bundles in the crista or neuromast. It is possible that the bundles are indeed thinner similar to Wang et al 2023.

Fig7A. IL1 should be indicated in this panel.

IL1 has been indicated, as suggested.

Fig7 supp 12. Color coding of the subunits would be appreciated here.

Done as requested.

Fig7. Overall the supplemental data for Figure 7 is quite extensive and the significance of this data is underappreciated. The authors could consider pushing panel C to supplemental as it is a second method to confirm the modeling interactions and instead highlight the dimer models which are more relevant than the monomer structures. Also, I find the additional alpha 0 helix quite interesting because it is not seen in the *C. elegans* cryoEM structure. Panel G should be given more importance instead of positioned deep into the figure next to the salt bridges in F. Overall, the novelty and significance of the modeling data deserves more importance in the paper.

We thank the reviewer for these helpful suggestions. The amphipathic alpha 0 helix is present in the *C. elegans* cryo-EM structure, although it is named differently in their paper (Jeong et al., 2022). We have now clarified this in the text: “Our new models feature an additional amphipathic helix, which we denote a0, extending almost parallel to the expected plane of the membrane bilayer without crossing towards the extracellular side (as observed for a mostly hydrophobic a0 in OSCA channels and labeled as H3 in the worm TMC-1 structure) …”. In addition, we have modified Figure 7 and highlighted panel G in a separate Figure 8 as requested.